

# Factors controlling the sequence of asperity failures in a fault model

Emanuele Lorenzano and Michele Dragoni

Dipartimento di Fisica e Astronomia, Alma Mater Studiorum Università di Bologna, Viale Carlo Berti Pichat 8, 40127 Bologna, Italy.

**Correspondence:** Emanuele Lorenzano (emanuele.lorenzano2@unibo.it)

**Abstract.** We consider a fault with two asperities embedded in a shear zone subject to a uniform strain rate owing to tectonic loading. The static stress field generated by seismic events undergoes viscoelastic relaxation as a consequence of the rheological properties of the asthenosphere. We treat the fault as a dynamical system whose basic elements are the asperities. The system has three degrees of freedom: the slip deficits of the asperities and the variation of their difference due to viscoelastic deformation. The dynamics of the system can be described in terms of one sticking mode and three slipping modes, for which we provide analytical solutions. We discuss how the stress state at the beginning of the interseismic interval preceding a seismic event controls the sequence of slipping modes during the event. We focus on the events associated with the separate (consecutive) slips of the asperities and investigate how they are affected by the seismic efficiency of the fault, by the difference in frictional resistance of the asperities and by the intensity of coupling between the asperities.

## 1 Introduction

Fault dynamics can be fruitfully investigated by asperity models (Lay et al., 1982; Scholz, 2002). In this framework, it is assumed that the fault plane is characterized by the presence of one or more strong regions with a high static friction and a velocity-weakening dynamic friction. As a consequence of tectonic loading, the stress acting on the asperities is gradually increased, eventually leading to their sudden failure and to a seismic event. Thus, asperity failures account for the unstable, stick-slip sliding regime of seismogenic faults. Examples of earthquakes that can be ascribed to the failure of two asperities are the 1964 Alaska earthquake (Christensen and Beck, 1994), the 2004 Parkfield, California, earthquake (Twardzik et al., 2012), the 2007 Pisco, Peru, earthquake (Sladen et al., 2010) and the 2010 Maule, Chile, earthquake (Delouis et al., 2010).

When considering asperity models, stress accumulation on the asperities, fault slip at the asperities and stress transfer between the asperities are factors of crucial relevance. It is therefore appropriate to describe the fault as a dynamical system whose essential components are the asperities (Ruff, 1992; Turcotte, 1997). The characterization through a finite number of degrees of freedom allows the study of the long-term evolution of the system by calculating its orbit in the phase space. Fur-





thermore, the dynamics can be described in terms of a finite number of modes, each one associated with a different system of ordinary differential equations. On the whole, it is possible to focus on the main features of the seismic source and avoid the more detailed and complicated characterization of continuum mechanics.

A two-asperity fault model was first considered by Nussbaum and Ruina (1987) and subsequently studied by Huang and Turcotte (1990), Rice (1993) and others. Further insights on the dynamics of a two-asperity fault were given by the introduction of a number of seismic features, such as post-seismic relaxation (Amendola and Dragoni, 2013), stress perturbations due to surrounding faults (Dragoni and Piombo, 2015) and elastic wave radiation (Dragoni and Santini, 2015). In these works, the fault was treated as a discrete dynamical system characterized by three slipping modes, corresponding to the failure of one or

both asperities at a time.

    Dragoni and Lorenzano (2015) considered a fault with two asperities of different strengths in the presence of viscoelastic relaxation. The authors showed that the knowledge of the state of the system at a given instant in time allows to calculate its orbit in the phase space and to predict its subsequent evolution, in the absence of stress perturbations. The orbit can be further

constrained from the observation of the earthquake source functions, which are related to the number and the sequence of slipping modes involved in the events. Accordingly, the uncertainty on the evolution of the system could be more and more reduced from the knowledge of the source functions of several consecutive earthquakes.

    The aim of the present paper is to expand the model of Dragoni and Lorenzano (2015) by including elastic wave radiation and

considering additional constraints on the state of the system during an interseismic phase. We solve analytically the equations of motion for each of the dynamic modes of the system. We discriminate the characteristics of a seismic event (number and sequence of slipping modes, seismic moment released, stress drops on the asperities) by identifying different subsets of states of the system. We focus on seismic events associated with the consecutive, but separate, slip of the asperities and discuss their relationship with the seismic efficiency of the fault. We retrieve additional constraints on the parameters of the system from the

knowledge of the stress states originating these kinds of events. We study how many phases of alternate slips of the asperities can be involved in an earthquake and show how this feature depends on the difference in frictional resistance of the asperities and on the intensity of coupling between the asperities.

## 2   The model

We consider a plane fault with two asperities of equal areas and different strengths, namely asperity 1 and asperity 2. The fault

is enclosed between two tectonic plates moving at constant relative velocity $V$ and embedded in a shear zone behaving like a homogeneous and isotropic Hooke solid. As a consequence of the relative motion of tectonic plates, the shear zone is subject to a uniform strain rate. We assume that coseismic stresses are relaxed with a characteristic Maxwell time $\Theta$, as a consequence of viscoelastic relaxation in the asthenosphere following an earthquake on the fault. Following Dragoni and Lorenzano (2015),





all quantities are expressed in nondimensional form.

Since asperities are characterized by a much higher friction than the surrounding region of the fault, we neglect the contribution of this weaker region to seismic moment. Instead of focusing on the values of friction, slip and stress at every point on
the fault, we only consider the average values of these quantities on each asperity.

We study the fault as a dynamical system with three state variables, functions of time $T$: the slip deficits $X(T)$ and $Y(T)$ of asperity 1 and asperity 2, respectively, and the variable $Z(T)$ representing the temporal variation of the difference between the slip deficits of the asperities, owing to viscoelastic relaxation in the asthenosphere. At a given instant in time, slip deficit
is defined as the slip that an asperity should undergo in order to recover the relative displacement of tectonic plates that took place up to that moment.

The tangential forces on the asperities (in units of the static friction on asperity 1) are

$$F_1 = -X + \alpha Z - \gamma \dot{X}, \qquad F_2 = -Y - \alpha Z - \gamma \dot{Y}. \tag{1}$$

In these expressions, the terms $-X$ and $-Y$ represent the effect of tectonic loading, whereas the terms $\pm \alpha Z$ correspond to the stress transfer between the asperities; finally, the terms $-\gamma \dot{X}$ and $-\gamma \dot{Y}$ are forces due to radiation damping during slip, where $\gamma$ is an impedance related with the seismic efficiency of the fault (Rice, 1993). The parameter $\alpha$ conveys the degree of coupling of the asperities.

As for friction on the asperities, we assume a simple rate-dependent law assigning a constant static friction and considering the average values of dynamic frictions during a slipping mode. This description of friction allows to replicate the typical stick-slip behaviour of fault dynamics. We assume that static friction on asperity 2 is a fraction $\beta$ of that on asperity 1 and that dynamic frictions are a fraction $\epsilon$ of static frictions.

A slip event takes place over a time interval very short with respect to the typical duration of interseismic intervals. Accordingly, viscoelastic relaxation can be reasonably neglected during a slip event and the equations of motion can be solved in the limit case of purely elastic coupling between asperities. This circumstance corresponds to (Amendola and Dragoni, 2013)

$$\Theta \to \infty, \qquad Z = Y - X. \tag{2}$$

Accordingly, during a slip event, the equations for the slip deficits $X$ and $Y$ are the same as in the case of purely elastic
coupling

$$\ddot{X} + \gamma \dot{X} + X - \alpha Z - \epsilon = 0 \tag{3}$$

$$\ddot{Y} + \gamma \dot{Y} + Y + \alpha Z - \beta \epsilon = 0 \tag{4}$$





while the variable $Z$ changes as

$$\ddot{Z} = \ddot{Y} - \ddot{X}. \tag{5}$$

The dynamics of the system can be characterized in terms of four dynamic modes: a sticking mode (00), corresponding to stationary asperities, and three slipping modes, corresponding to slip of asperity 1 alone (mode 10), slip of asperity 2 alone

(mode 01) and simultaneous slip of both asperities (mode 11). Each of these modes is associated with a specific system of autonomous ordinary differential equations.

In conclusion, the system is described by the six parameters $\alpha, \beta, \gamma, \epsilon, \Theta$ and $V$, subject to the constraints $\alpha \geq 0, 0 < \beta < 1, \gamma \geq 0, 0 < \epsilon < 1, \Theta > 0$ and $V > 0$.

## 10  2.1   The sticking region

During interseismic intervals of the fault, while both asperities are stationary (mode 00) and viscoelastic relaxation of coseismic stress takes place, the orbit of the system is enclosed in a particular subset of the state space $XYZ$. By definition, this subset corresponds to a phase of global stick of the system: accordingly, it is defined as the sticking region of the system (Di Bernardo et al., 2008). We show how it can be identified from the conditions for the occurrence of earthquakes on the fault and

a constraint on the state of stress of the fault.

During a global stick mode, the forces (1) reduce to

$$F_1 = -X + \alpha Z, \qquad F_2 = -Y - \alpha Z. \tag{6}$$

The conditions for the onset of motion for asperity 1 and 2 are, respectively,

$$F_1 = -1, \qquad F_2 = -\beta. \tag{7}$$

By combination with Eq. (6), we get the equations

$$X - \alpha Z - 1 = 0, \tag{8}$$

$$Y + \alpha Z - \beta = 0, \tag{9}$$

defining two planes in the $XYZ$ space, which we call $\Pi_1$ and $\Pi_2$, respectively.

We assume a condition of no overshooting: accordingly, we require that $X \geq 0$, $Y \geq 0$ and that the tangential forces on the asperities are always in the same direction as the velocity of tectonic plates, that is $F_1 \leq 0$, $F_2 \leq 0$. Again from Eq. (6), it is possible to define two additional planes in the $XYZ$ space,

$$X - \alpha Z = 0, \tag{10}$$





$$Y + \alpha Z = 0, \tag{11}$$

which we call $\Gamma_1$ and $\Gamma_2$, where $F_1 = 0$ and $F_2 = 0$, respectively.

To sum up, the sticking region is the subset of the $XYZ$ space enclosed by the planes $X = 0$, $Y = 0$, $\Pi_1$, $\Pi_2$, $\Gamma_1$ and $\Gamma_2$: a convex hexahedron **H** (Fig. 1). Accordingly, the sticking region **H** is a subset of the sticking region defined by Dragoni and Lorenzano (2015): in fact, they did not consider any constraint on the direction of the tangential forces on the asperities, so that the global stick phase of the fault was identified by a larger set of states. The vertices of **H** are the origin $(0,0,0)$ and the points

$$A = \left(0, 1, -\frac{1}{\alpha}\right), \; B = \left(\beta, 0, \frac{\beta}{\alpha}\right), \; C = \left(\beta + 1, 0, \frac{\beta}{\alpha}\right) \tag{12}$$

$$D = \left(0, \beta + 1, -\frac{1}{\alpha}\right), \; E = (1, 0, 0), \; F = (0, \beta, 0). \tag{13}$$

By definition, every orbit of mode 00 is enclosed within the sticking region and eventually reaches one of the faces $AECD$ or $BCDF$, belonging to the planes $\Pi_1$ and $\Pi_2$, respectively, giving rise to a seismic event. In these cases, the system enters mode

10 or mode 01, respectively. In the particular case in which the orbit of mode 00 reaches the edge $CD$, the system passes to mode 11.

    For later use, we introduce a point $P$ with coordinates

$$X_P = \frac{\alpha + \alpha\beta + 1}{1 + 2\alpha}, \qquad Y_P = \frac{\alpha + \alpha\beta + \beta}{1 + 2\alpha}, \qquad Z_P = -\frac{1 - \beta}{1 + 2\alpha} \tag{14}$$

belonging to the edge $CD$ and corresponding to a condition of purely elastic coupling, since $Z_P = Y_P - X_P$.

## 3   Solutions of dynamic modes

We solve the equations of motion for each of the four dynamic modes of the system. We shall make use of the frequencies

$$\omega_0 = \sqrt{1 - \frac{\gamma^2}{4}}, \qquad \omega_1 = \sqrt{1 + \alpha - \frac{\gamma^2}{4}}, \qquad \omega_2 = \sqrt{1 + 2\alpha - \frac{\gamma^2}{4}}. \tag{15}$$

We consider the case of underdamping, so that $\gamma \leq 2$: this choice is suggested by the observation that the seismic efficiency of

faults is small (Kanamori, 2001) and implies that the velocity dependent terms are small with respect to dynamic frictions. Let us define the slip amplitude of asperity 1 during a one-mode event 10 in the absence of radiation ($\gamma = 0$) as

$$U = 2\frac{1 - \epsilon}{1 + \alpha}. \tag{16}$$





Finally, we describe the effect of wave radiation by the quantity

$$\kappa = \frac{1}{2}\left(1 + e^{-\frac{\pi\gamma}{2\omega_1}}\right), \tag{17}$$

which is a decreasing function of $\gamma$, equal to 1 in the absence of radiation ($\gamma = 0$).

### 3.1 Stationary asperities (mode 00)

The variables $X$ and $Y$ increase steadily due to tectonic motion, while $Z$ is governed by the Maxwell constitutive equation. The equations of motion are

$$\ddot{X} = 0, \qquad \ddot{Y} = 0, \qquad \ddot{Z} = \frac{Z}{\Theta^2}, \tag{18}$$

where a dot indicates differentiation with respect to time $T$. Assuming an arbitrary initial state $(\bar{X}, \bar{Y}, \bar{Z}) \in \mathbf{H}$

$$X(0) = \bar{X}, \quad Y(0) = \bar{Y}, \quad Z(0) = \bar{Z} \tag{19}$$

and initial rates

$$\dot{X}(0) = V, \quad \dot{Y}(0) = V, \quad \dot{Z}(0) = -\frac{\bar{Z}}{\Theta} \tag{20}$$

the solution is

$$X(T) = \bar{X} + VT, \qquad Y(T) = \bar{Y} + VT, \qquad Z(T) = \bar{Z}e^{-T/\Theta,} \tag{21}$$

with $T \geq 0$. According to (21), during an interseismic interval the slip deficits of the asperities increase with time, as a result

of tectonic loading, while their difference undergoes viscoelastic relaxation.

We can retrieve the time $T_1$ required by the orbit of mode 00 to reach the plane $\Pi_1$ by imposing the condition

$$\bar{X} + VT_1 - \alpha\bar{Z}e^{-T_1/\Theta} - 1 = 0, \tag{22}$$

where we exploited Eq. (21). Accordingly, the slip of asperity 1 will start at

$$T_1 = \Theta W(\gamma_1) + \frac{1 - \bar{X}}{V}, \tag{23}$$

where $W$ is the Lambert function with argument

$$\gamma_1 = \frac{\alpha\bar{Z}}{V\Theta}e^{-\frac{1-\bar{X}}{V\Theta}}. \tag{24}$$

Analogously, the orbit of mode 00 intersects the plane $\Pi_2$ after a time $T_2$ satisfying the condition

$$\bar{Y} + VT_2 + \alpha\bar{Z}e^{-T_2/\Theta} - \beta = 0. \tag{25}$$





Thus, the slip of asperity 2 will start at

$$T_2 = \Theta W(\gamma_2) + \frac{\beta - \bar{Y}}{V}, \tag{26}$$

with

$$\gamma_2 = -\frac{\alpha \bar{Z}}{V\Theta} e^{-\frac{\beta - \bar{Y}}{V\Theta}}. \tag{27}$$

## 3.2 Slip of asperity 1 (mode 10)

The equations of motion are

$$\ddot{X} + \gamma \dot{X} + X - \alpha Z - \epsilon = 0 \tag{28}$$

$$\ddot{Y} = 0 \tag{29}$$

$$\ddot{Z} - \gamma \dot{X} - X + \alpha Z + \epsilon = 0. \tag{30}$$

The fault can enter mode 10 from mode 11 or from mode 00.

### 3.2.1 Case $11 \rightarrow 10$

After a phase of simultaneous motion, asperity 2 stops slipping and asperity 1 continues to slip alone. With initial conditions

$$X(0) = \bar{X}, \quad Y(0) = \bar{Y}, \quad Z(0) = \bar{Z} \tag{31}$$

$$\dot{X}(0) = \bar{V}, \quad \dot{Y}(0) = 0, \quad \dot{Z}(0) = -\bar{V} \tag{32}$$

the solution is

$$X(T) = \bar{X} - \frac{\bar{U}_1}{2} + \left[ \frac{\bar{U}_1}{2} \cos \omega_1 T + \frac{1}{\omega_1} \left( \frac{\gamma}{4} \bar{U}_1 + \bar{V} \right) \sin \omega_1 T \right] e^{-\frac{\gamma}{2}T} \tag{33}$$

$$Y(T) = \bar{Y} \tag{34}$$

$$Z(T) = \bar{Z} + \bar{X} - X(T) \tag{35}$$




where

$$\bar{U}_1 = 2\frac{\bar{X} - \alpha\bar{Z} - \epsilon}{1 + \alpha}. \tag{36}$$

If the orbit does not reach the plane $\Pi_2$ during the mode, asperity 1 stops slipping and the system goes back to a global stick phase; the slip duration can be calculated from the condition $\dot{X}(T) = 0$, yielding

$$T_{10} = \frac{1}{\omega_1}\left[\pi + \arctan\frac{2\omega_1\bar{V}}{(1 + \alpha)\bar{U}_1 + \gamma\bar{V}}\right]. \tag{37}$$

The final slip amplitude is then

$$U_{10} = \bar{X} - X(T_{10}) = \frac{\bar{U}_1}{2} + \sqrt{\frac{\bar{U}_1^2}{4} + \frac{\bar{V}^2}{1 + \alpha} + \frac{\gamma\bar{U}_1\bar{V}}{2(1 + \alpha)}}e^{-\frac{\gamma}{2}T_{10}}. \tag{38}$$

If instead the orbit reaches the plane $\Pi_2$ during the mode, the system enters again mode 11 and asperity 2 starts slipping together with asperity 1. The slip duration $T_{10}$ is then obtained by solving the equation

$$Y(T) + \alpha Z(T) - \beta = 0 \tag{39}$$

for the unknown $T$.

### 3.2.2   Case $00 \rightarrow 10$

Due to the combined effect of tectonic loading and viscoelastic relaxation, asperity 1 fails and starts slipping alone. In this case, the initial state belongs to the plane $\Pi_1$ given by Eq. (8): in fact, it is defined as the set of states where the condition for
the failure of asperity 1 is attained. Accordingly,

$$\bar{X} - \alpha\bar{Z} = 1, \qquad \bar{V} = 0 \tag{40}$$

and from Eq. (36)

$$\bar{U}_1 = U. \tag{41}$$

The solution reduces to

$$X(T) = \bar{X} - \frac{U}{2}\left[1 - \left(\cos\omega_1 T + \frac{\gamma}{2\omega_1}\sin\omega_1 T\right)e^{-\frac{\gamma}{2}T}\right] \tag{42}$$

$$Y(T) = \bar{Y} \tag{43}$$

$$Z(T) = \bar{Z} + \frac{U}{2}\left[1 - \left(\cos\omega_1 T + \frac{\gamma}{2\omega_1}\sin\omega_1 T\right)e^{-\frac{\gamma}{2}T}\right]. \tag{44}$$




If the orbit does not reach the plane $\Pi_2$ during the mode, asperity 1 stops slipping and the system goes back to a global stick phase; the slip duration and amplitude are, respectively,

$$T_{10} = \frac{\pi}{\omega_1}, \qquad U_{10} = \kappa U,$$
(45)

where $\kappa U$ is the maximum amount of slip of asperity 1 during mode 10.

If the orbit reaches the plane $\Pi_2$ before time $\pi/\omega_1$ has elapsed, the system passes to mode 11 and asperity 2 starts slipping together with asperity 1. In this case, the slip duration $T_{10}$ is obtained by solving Eq. (39) for the unknown $T$ with $Z(T)$ given by Eq. (44).

### 3.3 Slip of asperity 2 (mode 01)

The equations of motion are

$$\ddot{X} = 0$$
(46)

$$\ddot{Y} + \gamma\dot{Y} + Y + \alpha Z - \beta\epsilon = 0$$
(47)

$\ddot{Z} + \gamma\dot{Y} + Y + \alpha Z - \beta\epsilon = 0.$
(48)

The fault can enter mode 01 from mode 11 or from mode 00.

#### 3.3.1 Case 11 → 01

After a phase of simultaneous motion, asperity 1 stops slipping and asperity 2 continues to slip alone. With initial conditions

$$X(0) = \bar{X}, \quad Y(0) = \bar{Y}, \quad Z(0) = \bar{Z}$$
(49)

$$\dot{X}(0) = 0, \quad \dot{Y}(0) = \bar{V}, \quad \dot{Z}(0) = \bar{V}$$
(50)

the solution is

$$X(T) = \bar{X}$$
(51)

$Y(T) = \bar{Y} - \dfrac{\bar{U}_2}{2} + \left[\dfrac{\bar{U}_2}{2}\cos\omega_1 T + \dfrac{1}{\omega_1}\left(\dfrac{\gamma}{4}\bar{U}_2 + \bar{V}\right)\sin\omega_1 T\right]e^{-\frac{\gamma}{2}T}$
(52)





$$Z(T) = \bar{Z} - \bar{Y} + Y(T) \tag{53}$$

where

$$\bar{U}_2 = 2 \frac{\bar{Y} + \alpha \bar{Z} - \beta \epsilon}{1 + \alpha}. \tag{54}$$

If the orbit does not reach the plane $\Pi_1$ during the mode, asperity 2 stops slipping and the system goes back to a global stick phase; the slip duration can be calculated from the condition $\dot{Y}(T) = 0$, yielding

$$T_{01} = \frac{1}{\omega_1} \left[ \pi + \arctan \frac{2\omega_1 \bar{V}}{(1+\alpha)\bar{U}_2 + \gamma \bar{V}} \right]. \tag{55}$$

The final slip amplitude is then

$$U_{01} = \bar{Y} - Y(T_{01}) = \frac{\bar{U}_2}{2} + \sqrt{\frac{\bar{U}_2^2}{4} + \frac{\bar{V}^2}{1+\alpha} + \frac{\gamma \bar{U}_2 \bar{V}}{2(1+\alpha)}} e^{-\frac{\gamma}{2} T_{01}}. \tag{56}$$

If instead the orbit reaches the plane $\Pi_1$ during the mode, the system enters again mode 11 and asperity 1 starts slipping together with asperity 2. The slip duration $T_{01}$ is then obtained by solving the equation

$$X(T) - \alpha Z(T) - 1 = 0 \tag{57}$$

for the unknown $T$.

### 3.3.2   Case $00 \to 01$

As a result of the combined effect of tectonic loading and viscoelastic relaxation, asperity 2 fails and starts slipping alone. In this case, the initial state belongs to the plane $\Pi_2$ given by Eq. (9): in fact, it is defined as the set of states where the condition for the failure of asperity 2 is attained. Accordingly,

$$\bar{Y} + \alpha \bar{Z} = \beta, \quad \bar{V} = 0 \tag{58}$$

and from Eq. (54)

$$\bar{U}_2 = \beta U. \tag{59}$$

The solution reduces to

$$X(T) = \bar{X} \tag{60}$$

$$Y(T) = \bar{Y} - \frac{\beta U}{2} \left[ 1 - \left( \cos \omega_1 T + \frac{\gamma}{2\omega_1} \sin \omega_1 T \right) e^{-\frac{\gamma}{2} T} \right] \tag{61}$$

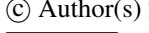
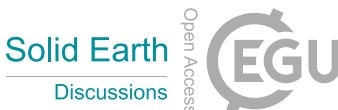

$$Z(T) = \bar{Z} - \frac{\beta U}{2} \left[ 1 - \left( \cos \omega_1 T + \frac{\gamma}{2\omega_1} \sin \omega_1 T \right) e^{-\frac{\gamma}{2}T} \right]. \tag{62}$$

If the orbit does not reach the plane $\Pi_1$ during the mode, asperity 2 stops slipping and the system goes back to a global stick phase; the slip duration and amplitude are, respectively,

$$T_{01} = \frac{\pi}{\omega_1}, \qquad U_{01} = \beta \kappa U, \tag{63}$$

where $\beta \kappa U$ is the maximum amount of slip of asperity 2 during mode 01.

If the orbit reaches the plane $\Pi_1$ before time $\pi/\omega_1$ has elapsed, the system passes to mode 11 and asperity 1 starts slipping together with asperity 2. In this case, the slip duration $T_{01}$ is obtained by solving Eq. (57) for the unknown $T$ with $Z(T)$ given by Eq. (62).

### 3.4 Simultaneous slip of asperities (mode 11)

The equations of motion are

$$\ddot{X} + \gamma \dot{X} + X - \alpha Z - \epsilon = 0 \tag{64}$$

$$\ddot{Y} + \gamma \dot{Y} + Y + \alpha Z - \beta \epsilon = 0 \tag{65}$$

$$\ddot{Z} + \gamma \left( \dot{Y} - \dot{X} \right) - X + Y + 2\alpha Z + (1 - \beta)\epsilon = 0 \tag{66}$$

and the solution is

$$X(T) = E_1 + (A \sin \omega_0 T + B \cos \omega_0 T + C \sin \omega_2 T + D \cos \omega_2 T) e^{-\frac{\gamma}{2}T} \tag{67}$$

$$Y(T) = E_2 + (A \sin \omega_0 T + B \cos \omega_0 T - C \sin \omega_2 T - D \cos \omega_2 T) e^{-\frac{\gamma}{2}T} \tag{68}$$

$$Z(T) = E_3 - 2 (C \sin \omega_2 T + D \cos \omega_2 T) e^{-\frac{\gamma}{2}T}, \tag{69}$$

where the constants $A$, $B$, $C$, $D$, $E_1$, $E_2$ and $E_3$ depend on initial conditions and are listed in Appendix A. The duration $T_{11}$ of mode 11 must be evaluated numerically: letting $T_x$ and $T_y$ be the smallest positive solutions of the equations $\dot{X}(T) = 0$ and $\dot{Y}(T) = 0$, respectively, we have $T_{11} = \min(T_x, T_y)$.





## 4 Relationship between stress states and slip episodes

In the framework of a two-asperity fault model, a seismic event is generally made up of $n$ slipping modes and can involve only one or both asperities at a time. More specifically, it is possible to distinguish three kinds of events, namely (i) events due to the slip of a single asperity, (ii) events associated with the consecutive, but separate, slips of both asperities and (iii)

events involving the simultaneous slip of asperities. The present model allows to gain information on the kind of seismic event generated by the fault from a geometrical point of view, each event being originated by a particular stress state corresponding to a specific subset of the state space. In the following, we first discuss the connection between the three kinds of events discussed above with the state of the system at the beginning of the earthquake. Afterwards, we show how the number and the sequence of slipping modes in a seismic event can be univocally determined from the knowledge of the state of the system at the beginning

of an interseismic interval, in the absence of stress perturbations.

### 4.1 Dependence on the state at the onset of the event

We showed in section 2 that the conditions for the onset of motion for asperity 1 and 2 are reached on the face $AECD$ and $BCDF$ of the sticking region $\mathbf{H}$, respectively. Here, we discuss the different subsets in which these faces can be divided, according to the number and sequence of dynamic modes involved in a seismic event. The purpose of this analysis is to point

out the relationship between the kind of seismic event generated by the fault and the state of the fault at the onset of the event itself.

Let us consider an orbit of mode 00 starting at a point $P_0$ inside $\mathbf{H}$ and reaching one of the faces $AECD$ or $BCDF$ at a point $P_k$, where the earthquake begins. With reference to Fig. 2, let us first focus on the face $AECD$. If $P_k$ belongs to the

trapezoid $\mathbf{Q_1}$, the earthquake will be a one-mode event 10; if $P_k$ belongs to the segment $\mathbf{s_1}$, the earthquake will be a two-mode event 10-01; finally, if $P_k$ belongs to the trapezoid $\mathbf{R_1}$, the earthquake will be a three-mode event 10-11-01 or 10-11-10. The specific sequence must be evaluated numerically and depends on the particular combination of the parameters $\alpha, \beta, \gamma$ and $\epsilon$. The remaining portion of the face would lead to overshooting. Analogous considerations can be made for subsets $\mathbf{Q_2}, \mathbf{s_2}$ and $\mathbf{R_2}$ on the face $BCDF$. In the particular case in which $P_k$ belongs to the edge $CD$, the earthquake will be a two-mode event 11-01.

There exists a correlation between the sequence of dynamic modes associated with the subsets of the faces $AECD$ and $BCDF$ and the distribution of forces on the fault. Let us consider an earthquake involving $n$ slipping modes starting with mode 10, i.e. on the face $AECD$. We call $P_i$ the representative point of the system at $T = T_i$, when the system enters the $i-$th mode $(i = 1, 2, ..., n)$. Finally, let $d$ be the distance of the starting point $P_1$ from the edge $CD$.

The magnitude $|F_2|$ of the force acting on asperity 2 at the beginning of the event $(T = T_1)$ decreases with $d$, as shown in Fig. 3(a), whereas the magnitude of the force $F_1$ acting on asperity 1 is the same everywhere $(|F_1| = 1)$. At $T = T_2$, the force on asperity 2 is

$$F_2(T_2) = F_2(T_1) - \alpha (Z_2 - Z_1), \tag{70}$$




owing to the stress transfer from asperity 1. If the magnitude of $F_2(T_1)$ is large enough that $|F_2(T_2)| = \beta$, the slip of asperity 1 triggers the slip of asperity 2, so that mode 10 is followed by mode 01 or 11. This condition is verified by states $P_1 \in \mathbf{s_1}$ and $P_1 \in \mathbf{R_1}$, respectively, as shown in Fig. 3(b); conversely, $|F_2(T_2)| < \beta$ for states $P_1 \in \mathbf{Q_1}$ and mode 10 is followed by mode 00.

Similar considerations hold for the face $BCDF$, with $|F_2| = \beta$ everywhere. This is shown in Fig. 4.

The boundaries of the subsets of the faces $AECD$ and $BCDF$ can be identified taking into account the no overshooting conditions and the constraint on the orientation of the tangential forces acting on the asperities discussed in section 2. The details are provided in Appendix B.

**4.2    Dependence on the state at the beginning of the interseismic interval**

We now discuss how the location of the initial point $P_0$ of any orbit of mode 00 affects the number and the sequence of slipping modes in the seismic event. Our aim is to illustrate how the kind of seismic event generated by the fault depends on the state of the fault at the beginning of the interseismic interval preceding the event itself.

Dragoni and Lorenzano (2015) showed the existence of a transcendental surface $\mathbf{\Sigma}$ which allows to discriminate the first slipping mode in a seismic event. In fact, this surface divides the sticking region $\mathbf{H}$ in two subsets $\mathbf{H_1}$ and $\mathbf{H_2}$. Given any initial state $P_0 \in \mathbf{H}$, the seismic event starts with mode 10 if $P_0 \in \mathbf{H_1}$ or with mode 01 if $P_0 \in \mathbf{H_2}$; in the particular case in which $P_0 \in \mathbf{\Sigma}$, the seismic event starts with mode 11. The surface $\mathbf{\Sigma}$ does not depend on the parameter $\gamma$; thus, it is not affected by seismic efficiency.

We now describe an additional surface inside each of the subsets $\mathbf{H_1}$ and $\mathbf{H_2}$, allowing to distinguish the number of slipping modes in a seismic event.

Let $P_1$ be the point where the orbit of mode 00 starting at $P_0 \in \mathbf{H_1}$ reaches the face $AECD$. In order that $P_1$ belongs to the
segment $\mathbf{s_1}$, its coordinates must satisfy Eq. (B4). Introducing the solutions (21) of mode 00 in Eq. (B4) and replacing $T_1$ with its expression (23), we obtain the equation of a transcendental surface $\mathbf{\Sigma_1}$

$$X - Y - 2\alpha Z e^{-\left(W(\gamma_1) + \frac{1-X}{\sqrt{v\Theta}}\right)} + \beta - \alpha\kappa U - 1 = 0, \tag{71}$$

where $W$ is the Lambert function with argument $\gamma_1$ defined in Eq. (24). The surface $\mathbf{\Sigma_1}$ is shown in Fig. 5. It lies beneath the surface $\mathbf{\Sigma}$, so that the subset $\mathbf{H_1}$ is divided into two sections $\mathbf{H_1^-}$ and $\mathbf{H_1^+}$, respectively below and above $\mathbf{\Sigma_1}$. If $P_0 \in \mathbf{H_1^-}$,
then $P_1 \in \mathbf{Q_1}$ and the earthquake will be a one-mode event, whereas if $P_0 \in \mathbf{H_1^+}$, then $P_1 \in \mathbf{R_1}$ and the earthquake will be a three-mode event, as discussed in the previous section. By definition, the segment $\mathbf{s_1}$ belongs to $\mathbf{\Sigma_1}$ and no orbit can cross $\mathbf{\Sigma_1}$: accordingly, if $P_0 \in \mathbf{\Sigma_1}$, its orbit remains on $\mathbf{\Sigma_1}$ and reaches the segment $\mathbf{s_1}$, giving rise to a two-mode event.



We now repeat the analysis for the subset $\mathbf{H_2}$. Let $P_2$ be the point where the orbit of mode 00 starting at $P_0 \in \mathbf{H_2}$ reaches the face $BCDF$. In order that $P_2$ belongs to the segment $\mathbf{s_2}$, its coordinates must satisfy Eq. (B11). Introducing the solutions (21) of mode 00 in Eq. (B11) and replacing $T_2$ with its expression (26), we obtain the equation of a transcendental surface $\mathbf{\Sigma_2}$

$$X - Y - 2\alpha Z e^{-\left(W(\gamma_2) + \frac{\beta - Y}{V\Theta}\right)} + \beta + \alpha\beta\kappa U - 1 = 0, \tag{72}$$

where the argument $\gamma_2$ has been defined in Eq. (27). The surface $\mathbf{\Sigma_2}$ is shown in Fig. 6. It lies above the surface $\mathbf{\Sigma}$, so that the subset $\mathbf{H_2}$ is divided into two sections $\mathbf{H_2^-}$ and $\mathbf{H_2^+}$, respectively below and above $\mathbf{\Sigma_2}$. If $P_0 \in \mathbf{H_2^-}$, then $P_2 \in \mathbf{R_2}$ and the earthquake will be a three-mode event, whereas if $P_0 \in \mathbf{H_2^+}$, then $P_2 \in \mathbf{Q_2}$ and the earthquake will be a one-mode event. By definition, the segment $\mathbf{s_2}$ belongs to $\mathbf{\Sigma_2}$ and no orbit can cross $\mathbf{\Sigma_2}$: accordingly, if $P_0 \in \mathbf{\Sigma_2}$, its orbit remains on $\mathbf{\Sigma_2}$ and reaches the segment $\mathbf{s_2}$, giving rise to a two-mode event.

In the purely elastic case, the surfaces $\mathbf{\Sigma_1}$ and $\mathbf{\Sigma_2}$ reduce to two lines in the $XY$ plane that were defined by Dragoni and Santini (2015). It is clear from their definitions (71) and (72) that both $\mathbf{\Sigma_1}$ and $\mathbf{\Sigma_2}$ depend on the maximum amount of slip allowed to asperity 1 in a one-mode event 10. Therefore, their position inside the sticking region changes as a function of $\gamma$. For larger values of $\gamma$, they are both closer to $\mathbf{\Sigma}$, so that the subsets $\mathbf{H_1^+}$ and $\mathbf{H_2^-}$ are smaller. This feature shows that higher values of $\gamma$ reduce the possibility of simultaneous slip of the asperities, in agreement with the results obtained by Dragoni and Santini (2015).

## 5  Seismic moment and stress drops on the asperities

The seismic moment released during an earthquake involving $n$ slipping modes can be retrieved from the knowledge of the total slip amplitudes of the asperities.

During the $i-$th mode, starting at time $T = T_i$ when the state of the system is $(X_i, Y_i, Z_i)$, the slips of asperity 1 and 2 are, respectively,

$$\Delta X_i = X_i - X_{i+1}, \qquad \Delta Y_i = Y_i - Y_{i+1} \tag{73}$$

with $i = 1, 2, ..., n$. The final slip amplitudes of asperity 1 and 2 are, respectively,

$$U_1 = \sum_{i=1}^{n} \Delta X_i = X_1 - X_{n+1}, \qquad U_2 = \sum_{i=1}^{n} \Delta Y_i = Y_1 - Y_{n+1}. \tag{74}$$

Accordingly, the final seismic moment is given by

$$M_0 = M_1 \frac{U_1 + U_2}{U}, \tag{75}$$

where $M_1$ is the seismic moment associated with a one-mode event 10 in the absence of wave radiation ($\gamma = 0$).





The slip rates of the asperities in an $n$-mode event are

$$\Delta \dot{X}(T) = \sum_{i=1}^{n} \Delta \dot{X}_i(T) \left[H(T - T_i) - H(T - T_{i+1})\right] \tag{76}$$

$$\Delta \dot{Y}(T) = \sum_{i=1}^{n} \Delta \dot{Y}_i(T) \left[H(T - T_i) - H(T - T_{i+1})\right] \tag{77}$$

where $H(T)$ is the Heaviside function. The moment rate of an $n$-mode event is then

$$\dot{M}(T) = M_1 \frac{\Delta \dot{X} + \Delta \dot{Y}}{U}. \tag{78}$$

Figures (7) and (8) show the evolution of the slip amplitude and the moment rate function associated with one-mode events 10 and 01, respectively, for a given choice of the parameters of the system.

The knowledge of the slip amplitudes of the asperities and the stress transferred from one asperity to the other allows to evaluate the static force drops on the asperities associated with the $n$-mode event. At the end of the earthquake, the static force drop on asperity 1 is

$$\Delta F_1 = F_1(T_{n+1}) - F_1(T_1) = U_1 + \alpha(Z_{n+1} - Z_1) \tag{79}$$

where we used the definitions of $F_1$ and $U_1$ given in Eq. (6) and Eq. (74), respectively. Analogously, the static force drop on
asperity 2 is

$$\Delta F_2 = F_2(T_{n+1}) - F_2(T_1) = U_2 - \alpha(Z_{n+1} - Z_1). \tag{80}$$

where we used the definitions of $F_2$ and $U_2$ given in Eq. (6) and Eq. (74), respectively.

The values of $M_0$, $\Delta F_1$ and $\Delta F_2$ can be discriminated according to the position of the point $P_1$ where the seismic event
starts, as summarized in Table 1. For events involving the slip of a single asperity, the force drop on the stationary asperity is negative, since stress is accumulated on it. The static stress drop on the asperities can be straightforwardly obtained dividing the static force drops by the area of the asperities.

## 6   Events due to the consecutive slip of the asperities

We focus on seismic events associated with the consecutive, but separate, slip of the asperities. First, we consider two-mode
events 10-01 and 01-10 and discuss how they are affected by the seismic efficiency of the fault. Afterwards, we exploit the knowledge of the stress states giving rise to such events in order to obtain additional constraints on the parameters of the system. Finally, we study how many phases of alternate slips of the asperities can be involved in an earthquake and how these particular sequences of dynamic modes are related to the parameters of the system.





## 6.1 Influence of the seismic efficiency

We illustrate how two-mode events 10-01 and 01-10 are affected by the radiation of elastic waves. To this aim, we study the effect of a variation of the parameter $\gamma$ in the interval $[0, 2]$. In the following, we shall use a superscript $^0$ when referring to quantities defined in the absence of wave radiation ($\gamma = 0$).

The lengths $l_1$ and $l_2$ of segments $\mathbf{s_1}$ and $\mathbf{s_2}$, respectively, as well as their distances $d_1$ and $d_2$ from the edge $CD$ are provided in Appendix B. In the limit case $\gamma = 0$, the maximum amount of slip $\kappa U$ of asperity 1 that is present in their expressions must be replaced by $U$ defined in Eq. (16), where $U \geq \kappa U$. In Fig. 9 we plot the ratios $l_1/l_1^0$ and $l_2/l_2^0$ as functions of $\gamma$. The trends clearly point out that an increase in $\gamma$ entails a lengthening of both segments $\mathbf{s_1}$ and $\mathbf{s_2}$. As a matter of fact, the lengths of

these segments depend on the coordinates of their end points, which are in turn constrained by the no overshooting conditions. Since wave radiation reduces the maximum amount of slip allowed to the asperities, the number of states satisfying the no overshooting conditions is increased and more states are included in the segments $\mathbf{s_1}$ and $\mathbf{s_2}$. As $\gamma$ grows, the probability that the system gives rise to a two-mode event 10-01 or 01-10 is thus enlarged.

According to Eq. (B15), the ratio $d_i/d_i^0$ is the same for both segments $\mathbf{s_1}$ and $\mathbf{s_2}$. It is shown in Fig. 10 as a function of $\gamma$. Evidently, an increase in $\gamma$ takes both segments $\mathbf{s_1}$ and $\mathbf{s_2}$ closer to the edge $CD$ of the sticking region. This can be explained if one considers the already discussed correlation between the different subsets of the faces $AECD$ and $BCDF$ and the forces acting on the asperities (section 4.1). Taking into account that wave radiation lowers the slip of the asperities, the stress transferred by one asperity to the other during a slip event is reduced as well. Thus, the segment $\mathbf{s_1}$ must be closer to the edge $CD$,

so that the value of $F_2$ at the beginning of mode 10 is large enough for the stress transferred by asperity 1 to asperity 2 to trigger mode 01. Analogous considerations can be made for the segment $\mathbf{s_2}$ on the face $BCDF$.

A direct consequence of the smaller distance between segments $\mathbf{s_1}$ and $\mathbf{s_2}$ and the edge $CD$ is that the areas $A_{\mathbf{Q_i}}$ of the subsets $\mathbf{Q_1}$ and $\mathbf{Q_2}$ are enlarged, while the areas $A_{\mathbf{R_i}}$ of the subsets $\mathbf{R_1}$ and $\mathbf{R_2}$ are reduced. This is shown in Fig. 11, where

we plot the ratios $A_{\mathbf{Q_i}}/A_{\mathbf{Q_i}}^0$ and $A_{\mathbf{R_i}}/A_{\mathbf{R_i}}^0$ as functions of $\gamma$. This feature provides an additional proof that higher seismic efficiency progressively reduces the possibility of simultaneous slip of the asperities.

## 6.2 Additional constraints on the parameters of the system

We introduced in section 2 the constraint $F_1 \leq 0$, $F_2 \leq 0$, requiring that the tangential forces on the asperities are always in the same direction as the velocity of tectonic plates. Accordingly, the ratio $F_1/F_2$ must always be a positive quantity. We

now exploit the knowledge of the particular stress states yielding to two-mode events 10-01 and 01-10 to establish additional constraints on the parameters of the system.





Let us first consider a two-mode event 10-01 taking place on the segment $\mathbf{s_1}$ on the face $AECD$ of the sticking region. Introducing the coordinates of any of the end points (B3)-(B5) of $\mathbf{s_1}$ in the expressions (6) of the forces acting on the asperities, we find that the stress state at the onset of the event is such that

$$\frac{F_1}{F_2} = \frac{1}{\beta - \alpha\kappa U}. \tag{81}$$

Imposing the condition $F_1/F_2 \geq 0$, we find

$$\beta \geq \alpha\kappa U. \tag{82}$$

Let us now focus on a two-mode event 01-10 taking place on the segment $\mathbf{s_2}$ on the face $BCDF$ of the sticking region. Introducing the coordinates of any of the end points (B10)-(B12) of $\mathbf{s_2}$ in the expressions (6) of the forces acting on the asperities, we find that the stress state at the onset of the event is such that

$$\frac{F_1}{F_2} = \frac{\beta}{1 - \alpha\beta\kappa U}. \tag{83}$$

Imposing the condition $F_1/F_2 \geq 0$, we get

$$\beta \leq \frac{1}{\alpha\kappa U}. \tag{84}$$

To sum up, the parameters of the system are subject to the condition

$$\alpha\kappa U \leq \beta \leq \frac{1}{\alpha\kappa U}. \tag{85}$$

Although these constraints have been obtained considering two particular seismic events, they represent a general feature of the present model.

### 6.3 Multiple consecutive slips

In the following, we investigate the conditions under which the system can generate a $n$-mode event involving the consecutive, but separate, slip of the asperities, with $n > 2$. To this aim, we recall that the slip deficit of asperity 1 is reduced by an amount

$\kappa U$ each time it slips alone; analogously, the slip deficit of asperity 2 is reduced by an amount $\beta\kappa U$ each time it slips alone.

#### 6.3.1 Three-mode events 10-01-10

At the end of a two-mode event 10-01, starting at a point $P_1 = (X_1, Y_1, Z_1)$ on the segment $\mathbf{s_1}$ on the face $AECD$ of the sticking region, the system is at a point $P_2$ with coordinates

$$X_2 = X_1 - \kappa U, \qquad Y_2 = Y_1 - \beta\kappa U, \qquad Z_2 = Z_1 + \kappa U(1 - \beta). \tag{86}$$

The event will then continue with a third mode 10 if $P_2 \in \Pi_1$: thus, introducing the coordinates of $P_2$ in Eq. (8) and bearing in mind that

$$X_1 = 1 + \alpha Z_1, \tag{87}$$





we get the following condition:

$$\alpha = \frac{1}{\beta - 1}. \tag{88}$$

As $0 < \beta < 1$, this result is unacceptable, since $\alpha$ is defined as positive. We conclude that, if we consider seismic events involving the alternate slips of the asperities, starting with the slip of asperity 1, the system can only generate a two-mode event

10-01. Any additional slip phase is prevented by the stronger frictional resistance of asperity 1 with respect to asperity 2.

### 6.3.2    Three-mode events 01-10-01

At the end of a two-mode event 01-10, starting at a point $P_1 = (X_1, Y_1, Z_1)$ on the segment $\mathbf{s_2}$ on the face $BCDF$ of the sticking region, the system is at a point $P_2$ with the same coordinates as given in Eq. (86). The event will then continue with a third mode 01 if $P_2 \in \Pi_2$: thus, introducing the coordinates of $P_2$ in Eq. (9) and bearing in mind that

$$Y_1 = \beta - \alpha Z_1, \tag{89}$$

we get the following condition:

$$\alpha = \alpha^* = \frac{\beta}{1 - \beta}. \tag{90}$$

Since $0 < \beta < 1$, the constraint $\alpha^* \geq 0$ is always satisfied. Accordingly, under the particular condition $\alpha = \alpha^*$, the system can give rise to three-mode events 01-10-01.

### 6.3.3    Four-mode events 01-10-01-10

At the end of a three-mode event 01-10-01, the system is at a point $P_3$ with coordinates

$$X_3 = X_1 - \kappa U, \qquad Y_3 = Y_1 - 2\beta\kappa U, \qquad Z_3 = Z_1 + \kappa U(1 - 2\beta). \tag{91}$$

The event will then continue with a fourth mode 10 if $P_3 \in \Pi_1$: thus, introducing the coordinates of $P_3$ in Eq. (8), bearing Eq. (B11) in mind and taking into account that $\alpha = \alpha^*$, we end up with

$$\beta = -1, \tag{92}$$

which is unacceptable, since $\beta$ is defined as positive. We conclude that, if we consider seismic events involving the alternate slips of the asperities, starting with the slip of asperity 2, the system can only generate two-mode events 01-10 and, under particular conditions related with the geometry of the fault and the coupling between the asperities, three-mode events 01-10-01.

To sum up, according to our analysis, the present model predicts $n-$mode events with $n \leq 3$; specifically, the sole seismic event involving three slipping modes (i.e., $n = 3$) is associated with the particular sequence 01-10-01, which can only take place under the condition (90). The existence of events involving more than three slipping modes in the framework of the present model may be object of future works.



## 7 Conclusions

We considered a plane fault embedded in a shear zone, subject to a uniform strain rate associated with tectonic loading. The fault contains two asperities of equal areas and different frictional resistance. After a seismic event, coseismic static stresses undergo viscoelastic relaxation.

We studied the fault as a dynamical system with three state variables: the slip deficits of the asperities and the variation of their difference in time, due to viscoelastic deformation. The sticking region of the system was identified in a convex hexahedron in the space of the state variables. We provided the analytical solutions to the equations of motion for the dynamics of the system, distinguishing between one sticking mode and three slipping modes. It was shown that events involving the

simultaneous slip of asperities take place from a small subset of states of the system.

We showed how the state of the system at the beginning of an interseismic interval constrains the state at the onset of the subsequent seismic event, and vice-versa. In turn, we related these details to the number and sequence of slipping modes involved in the earthquake, which determine the amount of seismic moment released and the stress drops on the asperities.

We discussed two-mode seismic events associated with the separate (consecutive) slips of the asperities, showing that they are favoured by higher seismic efficiencies. Taking advantage of the knowledge of the stress states generating these events, we retrieved further constraints on the parameters of the system. Finally, we showed that the model predicts the existence of three-mode events involving the separate slips of the asperities, provided that special conditions associated with the geometry

of the system are satisfied and that the event starts with the failure of the weaker asperity.

On the whole, the present work highlights an advantage granted by studying fault dynamics in the framework of discrete dynamical systems, where physical properties can be interpreted by and are related to geometrical features of the system (e.g. shape and dimension of different subsets of the state space, evolution of the orbit of the representative point of the system).

**Appendix A:  Constants in mode 11**

The fault can enter mode 11 from mode 10, 01 or 00. We list the constants $A$, $B$, $C$, $D$, $E_1$, $E_2$ and $E_3$ appearing in the solution for mode 11, discriminating between these three initial conditions.

**A1   Case $10 \rightarrow 11$**

The slip of asperity 1 triggers the failure of asperity 2, so that both asperities start slipping together. The initial conditions are

$$X(0) = \bar{X}, \quad Y(0) = \bar{Y}, \quad Z(0) = \bar{Z} \tag{A1}$$

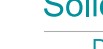
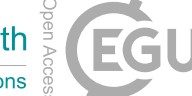


$$\dot{X}(0) = \bar{V}, \quad \dot{Y}(0) = 0, \quad \dot{Z}(0) = -\bar{V} \tag{A2}$$

with $\bar{Y}$ and $\bar{Z}$ satisfying the equation (9) of plane $\Pi_2$, which is defined as the set of states where the condition for the failure of asperity 2 is reached. The constants are

$$A = \frac{1}{2\omega_0} \left( \bar{V} + \gamma B \right) \tag{A3}$$

$$B = \frac{1}{2} \left[ \bar{X} + \bar{Y} - \epsilon \left( X_P + Y_P \right) \right] \tag{A4}$$

$$C = \frac{1}{2\omega_2} \left( \bar{V} + \gamma D \right) \tag{A5}$$

$$D = \frac{1}{2} \left( \epsilon Z_P + \frac{\bar{X} - \bar{Y} - 2\alpha \bar{Z}}{1 + 2\alpha} \right) \tag{A6}$$

$$E_1 = \epsilon X_P + \frac{\alpha}{1 + 2\alpha} \left( \bar{X} - \bar{Y} + \bar{Z} \right) \tag{A7}$$

$$E_2 = \epsilon Y_P - \frac{\alpha}{1 + 2\alpha} \left( \bar{X} - \bar{Y} + \bar{Z} \right) \tag{A8}$$

$$E_3 = \epsilon Z_P + \frac{1}{1 + 2\alpha} \left( \bar{X} - \bar{Y} + \bar{Z} \right). \tag{A9}$$

**A2  Case 01 → 11**

The slip of asperity 2 triggers the failure of asperity 1, so that both asperities start slipping together. The initial conditions are

$$X(0) = \bar{X}, \quad Y(0) = \bar{Y}, \quad Z(0) = \bar{Z} \tag{A10}$$

$$\dot{X}(0) = 0, \quad \dot{Y}(0) = \bar{V}, \quad \dot{Z}(0) = \bar{V} \tag{A11}$$

with $\bar{X}$ and $\bar{Z}$ satisfying the equation (8) of plane $\Pi_1$, which is defined as the set of states where the condition for the failure of asperity 1 is reached. The constants are

$$A = \frac{1}{2\omega_0} \left( \bar{V} + \gamma B \right) \tag{A12}$$





$$B = \frac{1}{2}\left[\bar{X} + \bar{Y} - \epsilon\left(X_P + Y_P\right)\right] \tag{A13}$$

$$C = \frac{1}{2\omega_2}\left(-\bar{V} + \gamma D\right) \tag{A14}$$

$$D = \frac{1}{2}\left(\epsilon Z_P + \frac{\bar{X} - \bar{Y} - 2\alpha\bar{Z}}{1 + 2\alpha}\right) \tag{A15}$$

$$E_1 = \epsilon X_P + \frac{\alpha}{1 + 2\alpha}\left(\bar{X} - \bar{Y} + \bar{Z}\right) \tag{A16}$$

$$E_2 = \epsilon Y_P - \frac{\alpha}{1 + 2\alpha}\left(\bar{X} - \bar{Y} + \bar{Z}\right) \tag{A17}$$

$$E_3 = \epsilon Z_P + \frac{1}{1 + 2\alpha}\left(\bar{X} - \bar{Y} + \bar{Z}\right). \tag{A18}$$

**A3    Case 00 → 11**

Owing to the combined effect of tectonic loading and viscoelastic relaxation, both asperities fail and start slipping together.

The initial conditions are

$$X(0) = \bar{X}, \quad Y(0) = \bar{Y}, \quad Z(0) = \bar{Z} \tag{A19}$$

$$\dot{X}(0) = 0, \quad \dot{Y}(0) = 0, \quad \dot{Z}(0) = 0 \tag{A20}$$

with $\bar{X}, \bar{Y}$ and $\bar{Z}$ satisfying both equations (8) and (9) of planes $\Pi_1$ and $\Pi_2$. The constants are

$$A = \frac{\gamma}{2\omega_0}B \tag{A21}$$

$$B = \frac{1 - \epsilon}{2}\left(X_P + Y_P\right) \tag{A22}$$

$$C = \frac{\gamma}{2\omega_2}D \tag{A23}$$



$$D = \frac{\epsilon - 1}{2} Z_P \tag{A24}$$

$$E_1 = \epsilon X_P + \frac{\alpha}{1 + 2\alpha} \left( \bar{X} - \bar{Y} + \bar{Z} \right) \tag{A25}$$

$$E_2 = \epsilon Y_P - \frac{\alpha}{1 + 2\alpha} \left( \bar{X} - \bar{Y} + \bar{Z} \right) \tag{A26}$$

$$E_3 = \epsilon Z_P + \frac{1}{1 + 2\alpha} \left( \bar{X} - \bar{Y} + \bar{Z} \right). \tag{A27}$$

## Appendix B: Details of the faces $AECD$ and $BCDF$

We provide here a description of the subsets of the faces $AECD$ and $BCDF$ of the sticking region $\mathbf{H}$ as they appear in Fig. 2.

Let us first focus on the face $AECD$. The vertices of the trapezoid $\mathbf{Q_1}$ are the point $E$ given in Eq. (13) and the points

$$G_1 = \left( \beta - \alpha \kappa U + 1, 0, \frac{\beta - \alpha \kappa U}{\alpha} \right) \tag{B1}$$

$$K_1 = \left( \kappa U, 1 - \kappa U, \frac{\kappa U - 1}{\alpha} \right) \tag{B2}$$

$$I_1 = \left( \kappa U, \beta - \kappa U (1 + \alpha) + 1, \frac{\kappa U - 1}{\alpha} \right). \tag{B3}$$

The segment $\mathbf{s_1}$ lies on the line

$$\begin{cases} X + Y - \beta + \alpha \kappa U - 1 = 0 \\ X - \alpha Z - 1 = 0 \end{cases} \tag{B4}$$

and its end points are the points $I_1$ and

$$H_1 = \left( \beta - \kappa U (\alpha + \beta) + 1, \beta \kappa U, \frac{\beta - \kappa U (\alpha + \beta)}{\alpha} \right). \tag{B5}$$

The vertices of the trapezoid $\mathbf{R_1}$ are the end points of $\mathbf{s_1}$ and the points

$$J_1 = \left( \kappa U, \beta - \kappa U + 1, \frac{\kappa U - 1}{\alpha} \right) \tag{B6}$$




$$J_2 = \left( \beta\left(1 - \kappa U\right) + 1, \beta \kappa U, \frac{\beta\left(1 - \kappa U\right)}{\alpha} \right). \tag{B7}$$

We now consider the face $BCDF$. The vertices of the trapezoid $\mathbf{Q_2}$ are the point $F$ given in Eq. (13) and the points

$$G_2 = \left( 0, \beta\left(1 - \alpha\kappa U\right) + 1, \beta\kappa U - \frac{1}{\alpha} \right) \tag{B8}$$

$$K_2 = \left( \beta\left(1 - \kappa U\right), \beta\kappa U, \frac{\beta\left(1 - \kappa U\right)}{\alpha} \right) \tag{B9}$$

$$I_2 = \left( \beta\left(1 - \kappa U - \alpha\kappa U\right) + 1, \beta\kappa U, \frac{\beta\left(1 - \kappa U\right)}{\alpha} \right). \tag{B10}$$

The segment $\mathbf{s_2}$ lies on the line

$$\begin{cases} X + Y - \beta\left(1 - \alpha\kappa U\right) - 1 = 0 \\ Y + \alpha Z - \beta = 0 \end{cases} \tag{B11}$$

and its end points are the points $I_2$ and

$$H_2 = \left( \kappa U, \beta - \kappa U(\alpha\beta + 1) + 1, \frac{\kappa U(\alpha\beta + 1) - 1}{\alpha} \right). \tag{B12}$$

The vertices of the trapezoid $\mathbf{R_2}$ are the end points of $\mathbf{s_2}$ and the points $J_1$ and $J_2$.

The lengths of segments $\mathbf{s_1}$ and $\mathbf{s_2}$ are, respectively,

$$l_1 = |\beta + 1 - \kappa U(1 + \alpha + \beta)| \sqrt{\frac{1 + 2\alpha^2}{\alpha^2}} \tag{B13}$$

$$l_2 = |\beta + 1 - \kappa U(1 + \alpha\beta + \beta)| \sqrt{\frac{1 + 2\alpha^2}{\alpha^2}}. \tag{B14}$$

The distances of segments $\mathbf{s_1}$ and $\mathbf{s_2}$ from the edge $CD$ are, respectively,

$$d_1 = \alpha\kappa U \sqrt{\frac{1 + \alpha^2}{1 + 2\alpha^2}}, \qquad d_2 = \beta d_1. \tag{B15}$$

*Author contributions.* E. L. developed the model, produced the figures and wrote a preliminary version of the paper; M. D. checked the equations and revised the text. Both authors discussed extensively the results.

*Competing interests.* The authors declare that they have no conflict of interest.



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





### List of Figure Captions

**Fig. 1** The sticking region of the system: a convex hexahedron $\mathbf{H}$ ($\alpha = 1, \beta = 1$). The point $P$, corresponding to purely elastic coupling between the asperities, is shown. Seismic events take place on the faces $AECD$ and $BCDF$

**Fig. 2** The faces $AECD$ and $BCDF$ of the sticking region and their subsets, which determine the number and the sequence of dynamic modes during a seismic event ($\alpha = 1, \beta = 1, \epsilon = 0.7$). The events taking place on the face $AECD\,(BCDF)$ start with mode $10\,(01)$

**Fig. 3** Force $F_2$ on asperity 2 during an earthquake involving $n$ slipping modes and starting with mode 10, as a function of the distance $d$ of the initial state $P_1$, measured on the face $AECD$ from the edge $CD$ of the sticking region $\mathbf{H}$ ($\alpha = 1$, $\beta = 0.5, \gamma = 1, \epsilon = 0.7$) : $(a)$ magnitude of $F_2$ at the onset of the event ($T = T_1$); $(b)$ magnitude of $F_2$ after the initial slip of asperity 1 ($T = T_2$). The labels indicate the subsets of the face $AECD$ corresponding to different intervals of $d$. The dashed line indicates the condition for the slip of asperity 2 ($|F_2| = \beta$), which is reached only for states $P_1 \in \mathbf{s_1}$ and $P_1 \in \mathbf{R_1}$

**Fig. 4** Force $F_1$ on asperity 1 during an earthquake involving $n$ slipping modes and starting with mode 01, as a function of the distance $d$ of the initial state $P_1$, measured on the face $BCDF$ from the edge $CD$ of the sticking region $\mathbf{H}$ ($\alpha = 1$, $\beta = 0.5, \gamma = 1, \epsilon = 0.7$) : $(a)$ magnitude of $F_1$ at the onset of the event ($T = T_1$); $(b)$ magnitude of $F_1$ after the initial slip of asperity 2 ($T = T_2$). The labels indicate the subsets of the face $BCDF$ corresponding to different intervals of $d$. The dashed line indicates the condition for the slip of asperity 1 ($|F_1| = 1$), which is reached only for states $P_1 \in \mathbf{s_2}$ and $P_1 \in \mathbf{R_2}$

**Fig. 5** The surface $\mathbf{\Sigma_1}$ in the subset $\mathbf{H_1}$ of the sticking region, discriminating the number of slipping modes in a seismic event starting when the orbit of the system reaches the face $AECD$ ($\alpha = 1, \beta = 1, \gamma = 1, \epsilon = 0.7, V\Theta = 1$)

**Fig. 6** The surface $\mathbf{\Sigma_2}$ in the subset $\mathbf{H_2}$ of the sticking region, discriminating the number of slipping modes in a seismic event starting when the orbit of the system reaches the face $BCDF$ ($\alpha = 1, \beta = 1, \gamma = 1, \epsilon = 0.7, V\Theta = 1$)

**Fig. 7** $(a)$ Slip amplitude and $(b)$ moment rate function associated with a one-mode event 10 ($\alpha = 1, \gamma = 1, \epsilon = 0.7$)

**Fig. 8** $(a)$ Slip amplitude and $(b)$ moment rate function associated with a one-mode event 01 ($\alpha = 1, \beta = 0.5, \gamma = 1, \epsilon = 0.7$)

**Fig. 9** The lengths $l_1/l_1^0$ and $l_2/l_2^0$ of segments $\mathbf{s_1}$ and $\mathbf{s_2}$ as functions of $\gamma$ ($\alpha = 1, \beta = 0.5, \epsilon = 0.7$). Larger values of the ratios $l_i/l_i^0$ entail a higher probability of a two-mode event associated with the separate slip of both asperities

**Fig. 10** The distance $d/d^0$ of segments $\mathbf{s_1}$ and $\mathbf{s_2}$ from the edge $CD$ as a function of $\gamma$ ($\alpha = 1, \epsilon = 0.7$). The smaller the distance, the more homogeneous the stress distribution on the fault at the beginning of a two-mode event associated with the separate slip of both asperities

**Fig. 11** The areas $A_{\mathbf{Q_i}}/A_{\mathbf{Q_i}}^0$ and $A_{\mathbf{R_i}}/A_{\mathbf{R_i}}^0$ of the subsets $\mathbf{Q_1}, \mathbf{Q_2}, \mathbf{R_1}$ and $\mathbf{R_2}$ as functions of $\gamma$ ($\alpha = 1, \beta = 0.5, \epsilon = 0.7$). As the ratios $A_{\mathbf{Q_i}}/A_{\mathbf{Q_i}}^0$ increase, the possibility of simultaneous slip of asperities is reduced. The converse holds for the ratios $A_{\mathbf{R_i}}/A_{\mathbf{R_i}}^0$





# Figures

**Figure 1.**

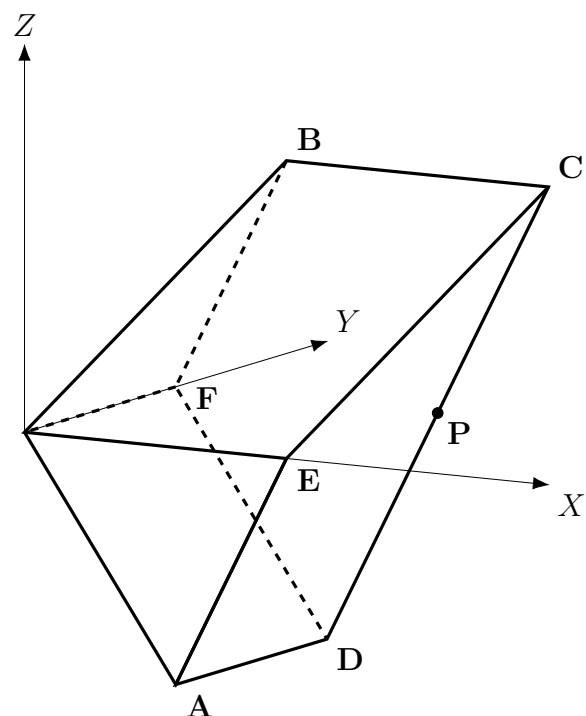



**Figure 2.**

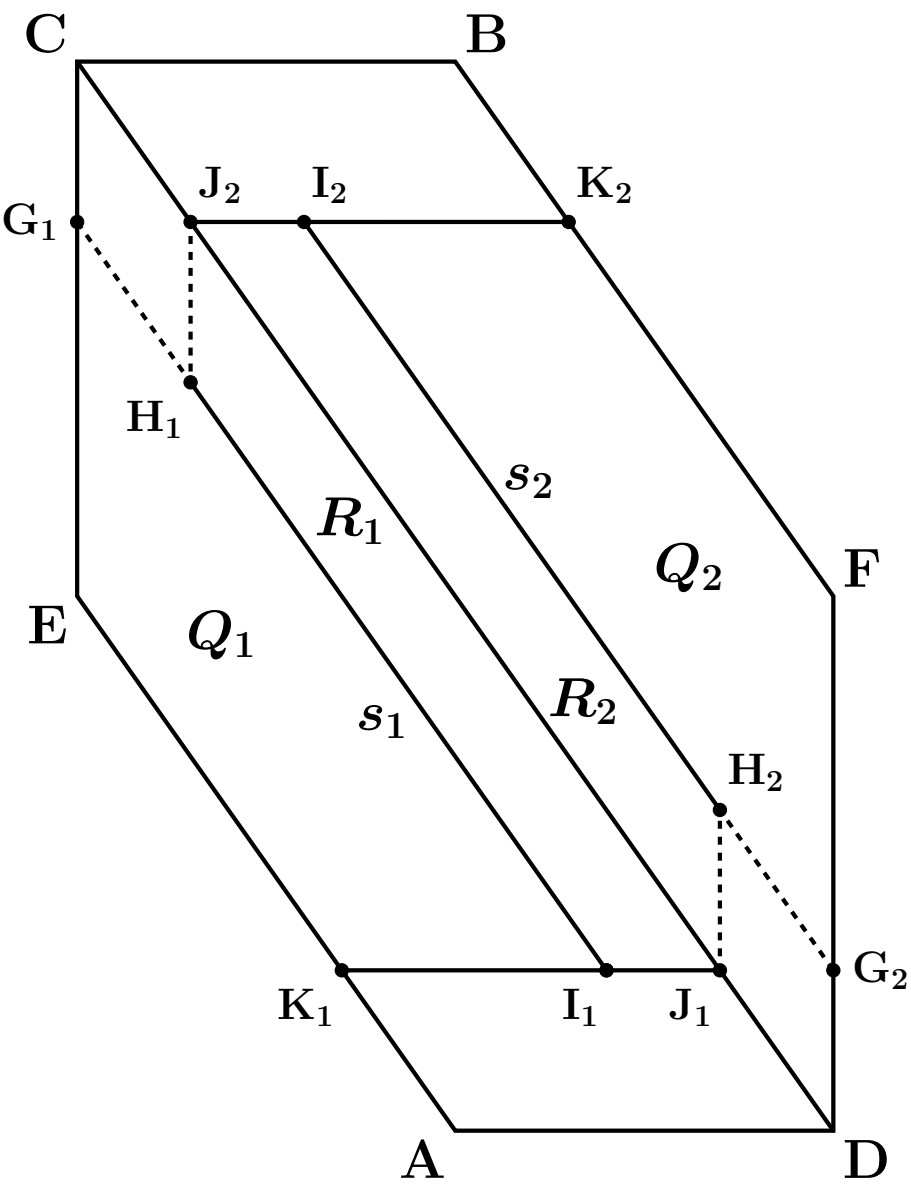



**Figure 3.**

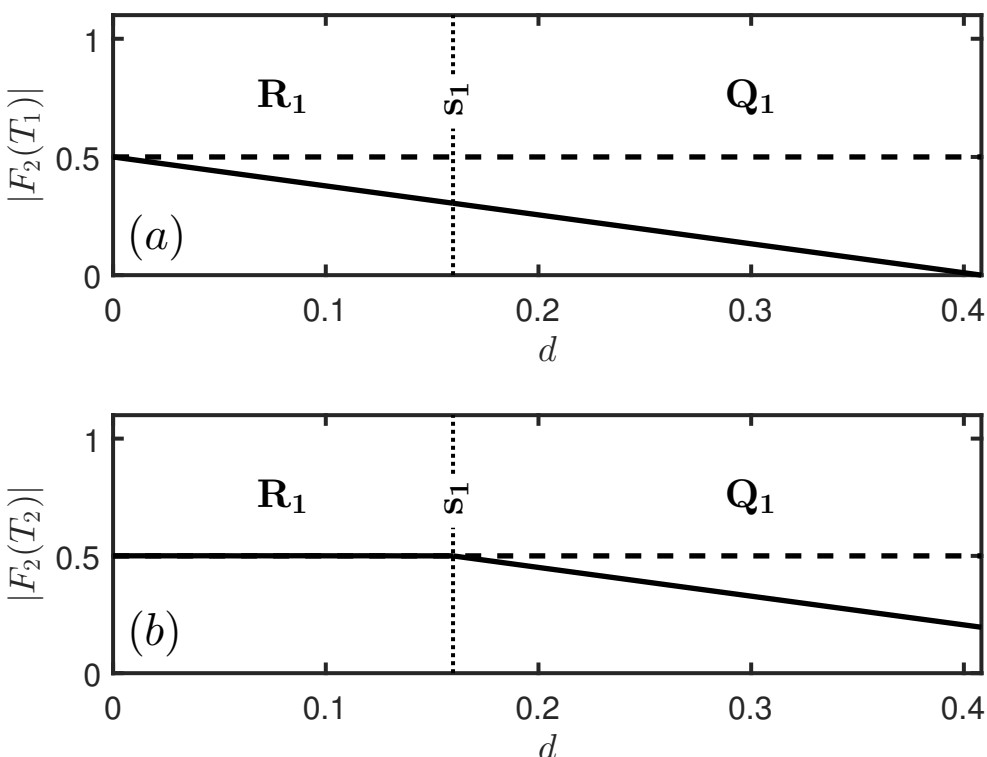



**Figure 4.**

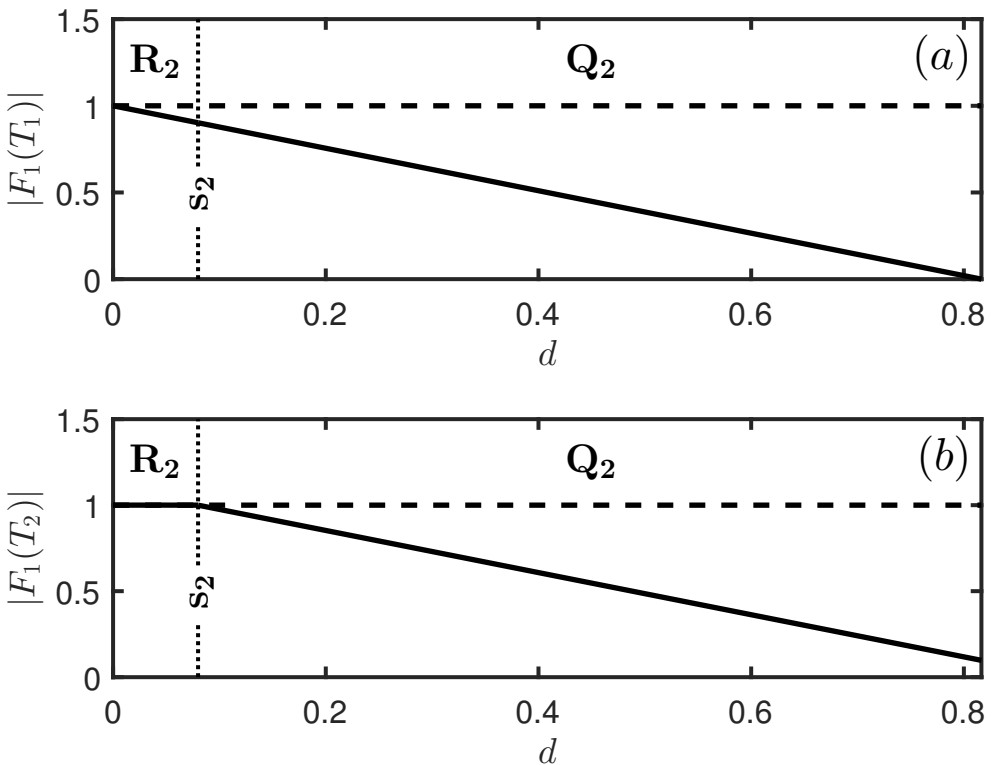





**Figure 5.**

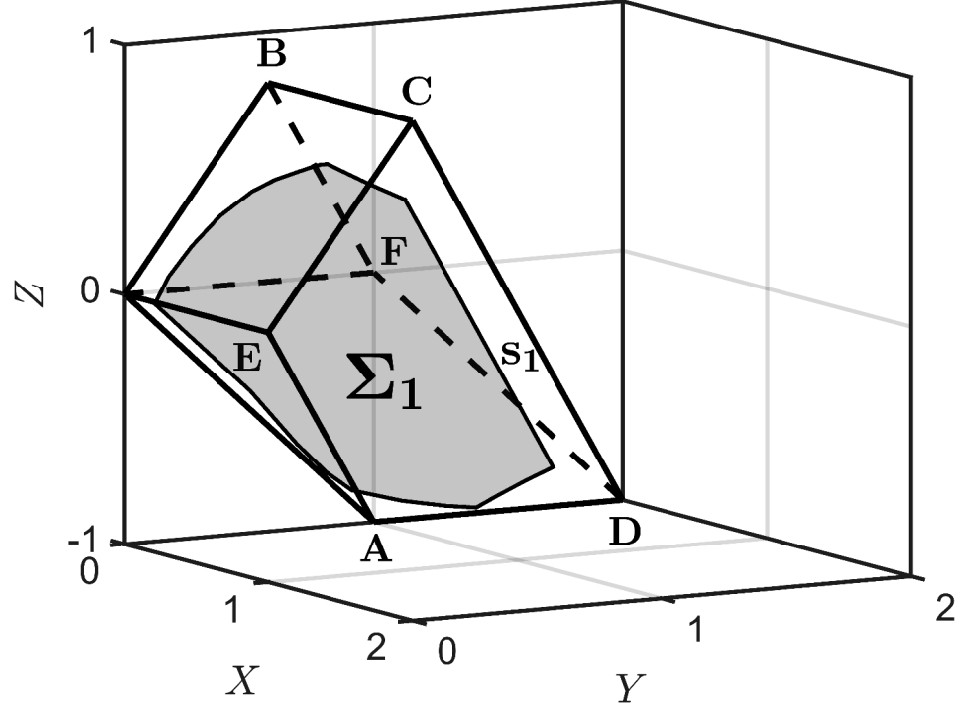



**Figure 6.**

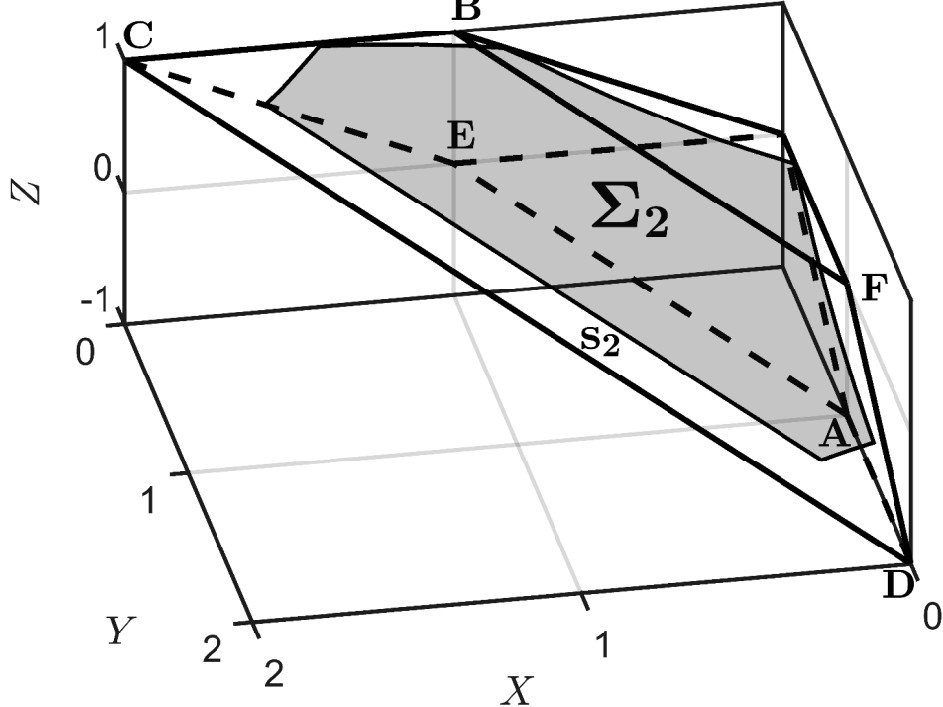



**Figure 7.**

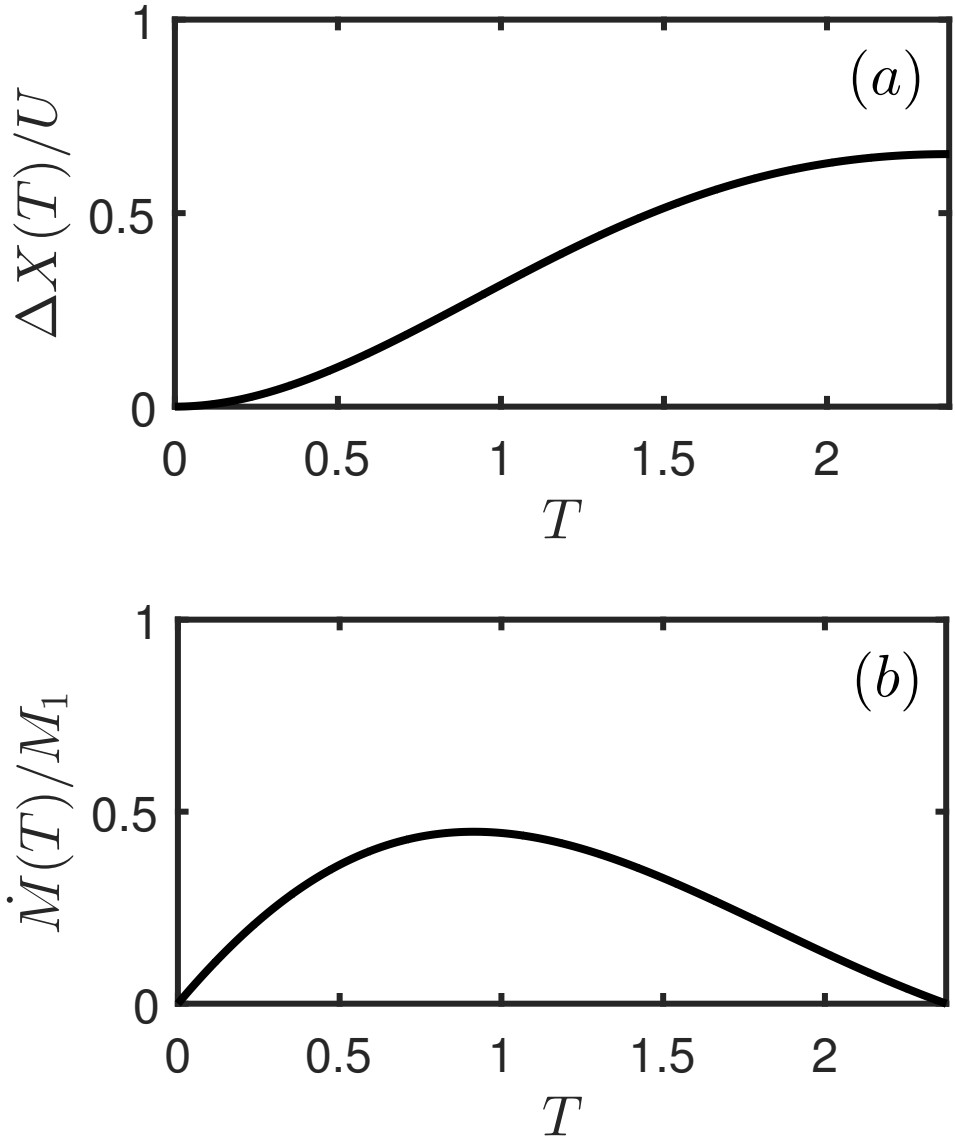





**Figure 8.**

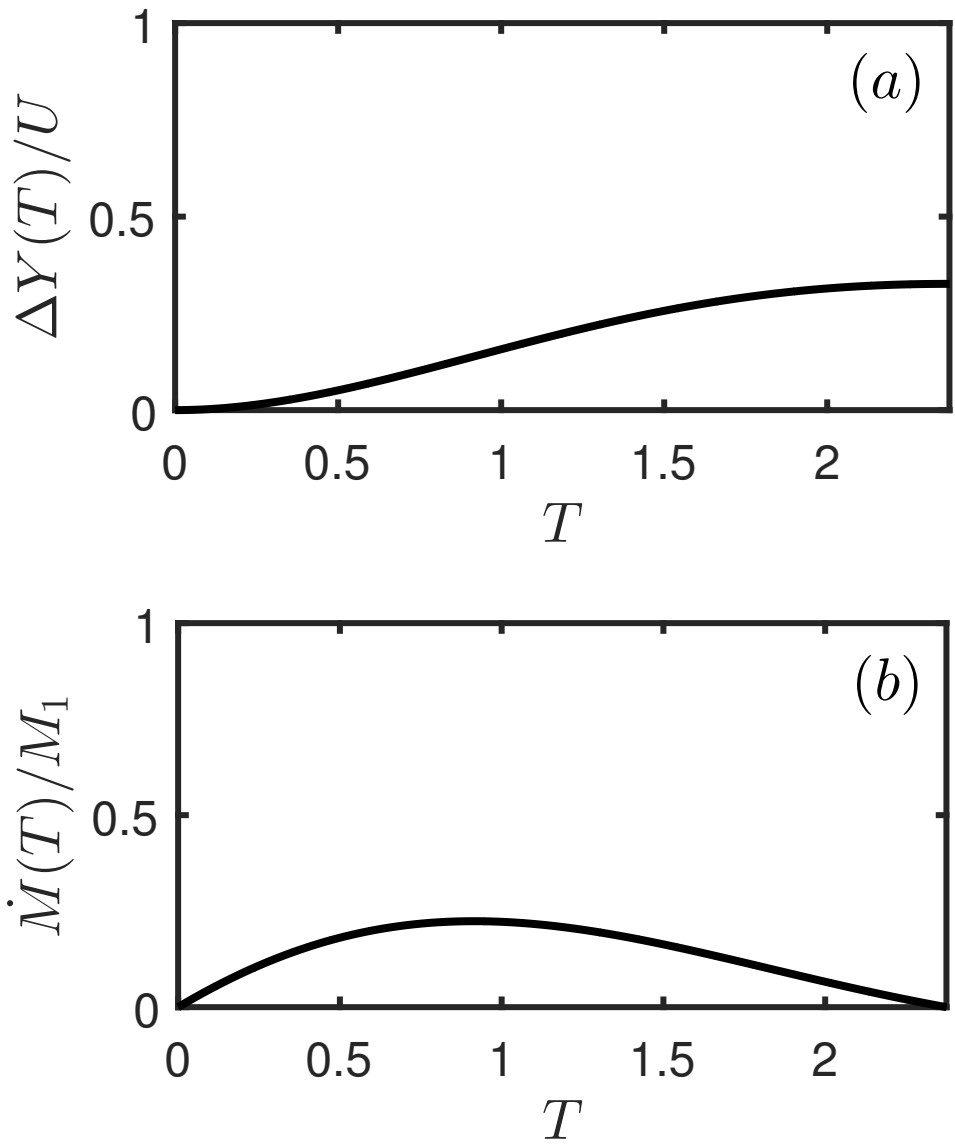





**Figure 9.**

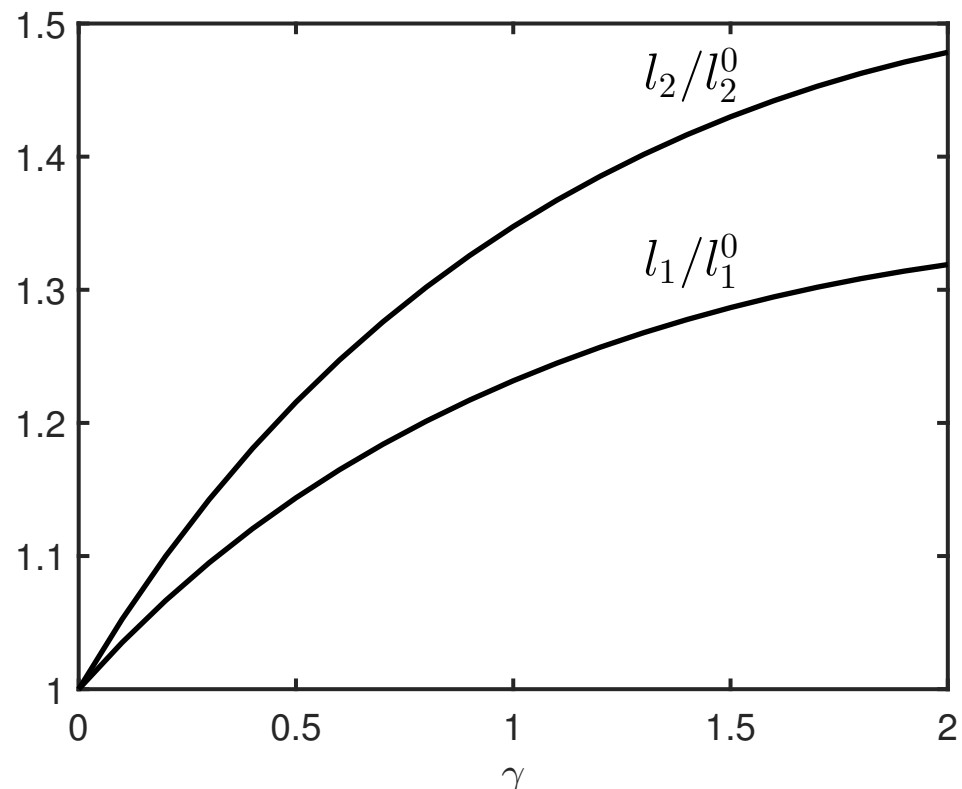





**Figure 10.**

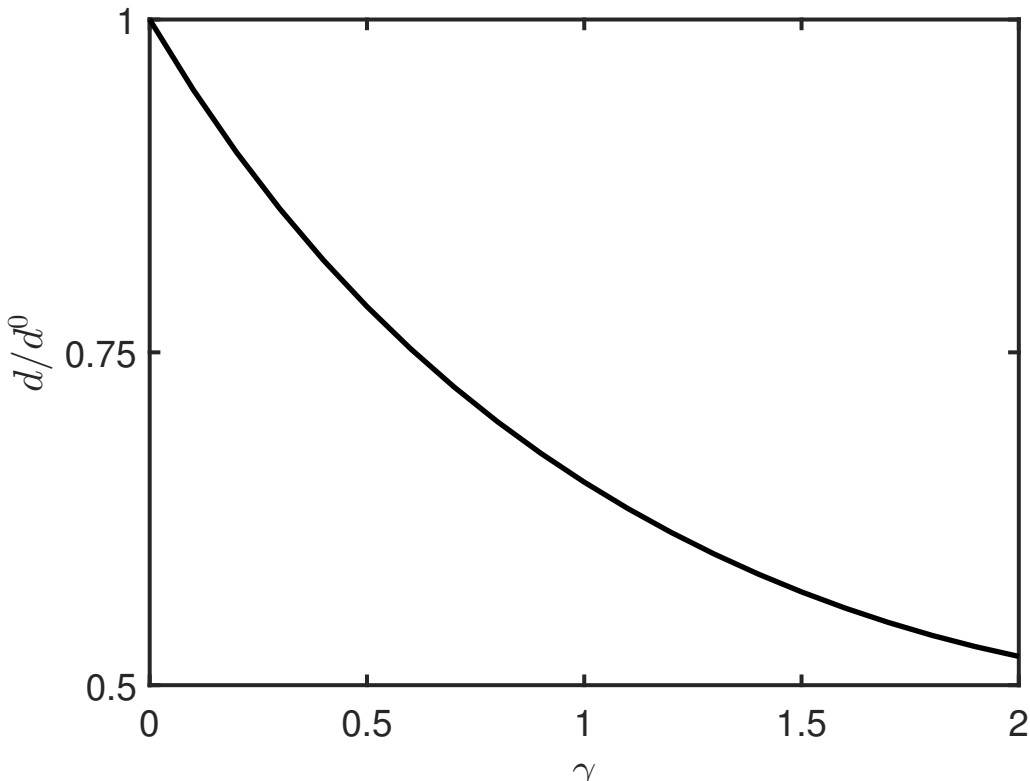





**Figure 11.**

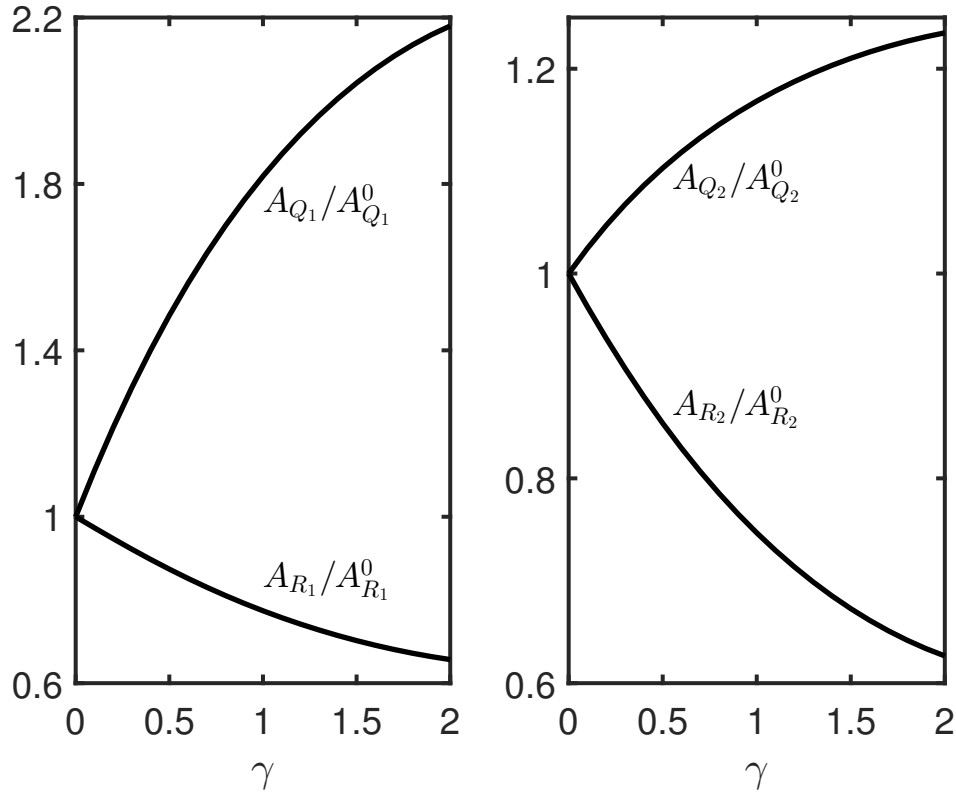





# Tables

**Table 1.** Final seismic moment $M_0$ and static force drops $\Delta F_1$, $\Delta F_2$ on asperity 1 and 2 following an earthquake involving $n$ slipping modes, as a function of the state $P_1$ where the event started. The entry *e.n.* is the abbreviation for *evaluated numerically*.

| State $P_1$ | $n$ | $M_0$ | $\Delta F_1$ | $\Delta F_2$ |
|---|---|---|---|---|
| $P_1 \in \mathbf{Q_1}$ | 1 | $\kappa M_1$ | $2\kappa(1-\epsilon)$ | $-\alpha\kappa U$ |
| $P_1 \in \mathbf{Q_2}$ | 1 | $\beta\kappa M_1$ | $-\alpha\beta\kappa U$ | $2\kappa\beta(1-\epsilon)$ |
| $P_1 \in \mathbf{s_1} \vee P_1 \in \mathbf{s_2}$ | 2 | $\kappa M_1(1+\beta)$ | $\kappa U(1+\alpha-\alpha\beta)$ | $\kappa U(\beta-\alpha+\alpha\beta)$ |
| $P_1 \in \mathbf{R_1} \vee P_1 \in \mathbf{R_2}$ | 3 | *e.n.* | *e.n.* | *e.n.* |