# Peer review of "Factors controlling the sequence of asperity failures in a fault model"

_Solid Earth, 2018_

## Referee Comment (RC1) · Anonymous Referee #1 · 4 Jun 2018

The presented manuscript aims to investigate the factors that control fault rupture sequences in a scenario where a fault is made up of two asperities. These two asperities are thought to be present along a fault "plane" (shear zone) that is otherwise freely slipping (i.e., does not build up strain). Additionally to elastic interaction between those asperities, the authors have added viscoelastic relaxation to approximate the post-seismic behavior of the coseismically strained asthenosphere. The authors treat their investigation in a formalized mathematical framework –a dynamical system whose basic elements are the aforementioned asperities. Although I am not the right reviewer to evaluate the mathematical formulation of the setup that is used, I still want to provide some comments.

General comments: Generally speaking, I find the treatment of the "earthquake"system

overly simplified –to the degree that I want to question whether the provided results actually bear any insights into the recurrence of earthquake rupture (including the effect of viscoelastic relaxation).

The authors mention that knowledge of the initial state of stress in the system would allow to calculate/predict the following sequence of earthquakes i.e., asperity ruptures (in absence of stress perturbations). While this may be in theory correct, this approach is in my view not appropriate to describe earthquake rupture and recurrence, considering the spatial and temporal variation of physical parameters that in fact control earthquake rupture. In the present work, all that existing and important complexity is removed i.e., not considered.

The authors mention that the aim of the presented study is to expand on previous work (P2L19). But that is not really motivating anything. What are the authors actually trying to constrain/identify? How can the results applied? What insights regarding earthquake rupture does it provide? The study needs an improved motivation/introduction section.

The proximity of the two asperities considered here relative to each other should play a role (on the probability of respective rupture modes) -maybe I missed it, but do the authors consider that point?

Below are by-line comments:

By line: P1L4 –colon after "degrees of freedom" indicates that a list of those is following –but that does not seem to be the case; please rephrase

P1L5 –the slipping modes should be mentioned here; current formulation too implicit/vague

Abstract – does not stand alone; the reader learns to some extend what the authors wanted to do but not what they learned/have found out; this needs to be included into the abstract

P1L13 –replace "by asperity models " -> the "by" is wrong -> the models don't investigate anything

P1L22 –how are "non-asperties" defined/characterized? Needs to be mentioned here; they also have a role within the earthquake system and the authors need to state what that role is; include corresponding explanation in the model formulation section.

P2L17 –what the authors mean with "source functions"? is that source time function? Please clarify

P2L24 –the term "seismic efficiency" should be defined properly

P3L3 – language is vague "by a much higher friction than the surrounding region of the fault" -> be specific/quantitative please

P3L4 –I cannot follow that logic: the authors argue that they can neglect the seismic moment contributed from the "weaker regions" of the fault that surround the asperities -> but why? Regardless of strength, if the fault slips (coseismically) then it will contribute to seismic moment -> so I want to question the author's approach here; they need to better explain justify this simplification.

P3L20 –why using a rate-dependent law? Did the authors experiment with other laws as well? Please better motivate the use of this friction law.

P12L8 –the authors will need to explain how their analytical toy model is able to inform our understanding of earthquake rupture and rupture sequences; after all, we don't know the "initial state of stress" and real faults do exhibit stress perturbations, along with a range of other processes and parameters that affect earthquake rupture and that are not considered here. So, how does the presented study help to learn about earthquakes?

---

## Referee Comment (RC2) · Anonymous Referee #2 · 18 Jun 2018

This paper solves analytically the modes of a fault with two asperities and discusses how the source processes are affected by seismic efficiency, frictional resistance and the intensity of coupling. It is written in a logical way and provides a different perspective on modeling earthquake source process. However, I think the authors need to work on how their approach relates to other modeling approaches and how it can be applied to realistic cases. I have outlined my major comments below:

1. Previous models have studied fully dynamic earthquake cycles on a fault with asperities [e.g., Lui and Lapusta, "Repeating microearthquake sequenes interact predominantly through postseismic slip", Nature Communications, 2016]. A review of these previous studies is lacking in the introduction section. In particular, how can the modeling approach in this paper contribute to our understanding of earthquake cycles?

2. On Page 3, the asperity is characterized by a much higher friction than the surrounding region, which I don't think is necessarily true for a real fault. Could the authors provide some observations that support this view?

3. The model assumes a rate-dependent friction law instead of a rate and state dependent friction law that is observed in laboratory experiments and used in fully dynamic earthquake cycle models. The authors replied to the other reviewer that using rate and state dependent friction laws would "provide negligible improvements to our conclusions". However, if the friction changes over time as defined by the state variable, it will significantly affect the recurrence intervals of seismic events.

4. On Page 5, the authors mentioned that they consider the case of underdamping because seismic efficiency of faults is small. I don't think this is true. Radiation efficiency depends on the earthquake type and is not always small. I've attached Figure 8 from Venkataramen and Kanamori [2004]. For example, the radiation efficiency of tsunami earthquakes is usually lower than other types of earthquakes.

5. It's hard to relate the proposed models to realistic cases. In section 4.1, the authors discussed the different earthquake models when Pk belongs to different segments on the face AECD. If we picked a region, e.g., Parkfield, how could we determine which segment it belongs to?

6. Fig.7 and Fig. 8 are not cited in the manuscript. Though the peak moment rate amplitudes are slightly different in the figures, the moment rate functions have very similar shapes for events 10 and 01. Why is that?

[Figure]

**Figure 8.** Radiation efficiencies determined from the radiated energy-to-moment ratios are plotted as a function of moment magnitude. The different symbols show different types of earthquakes as described in the legend. Most earthquakes have radiation efficiencies greater than 0.25, but tsunami earthquakes and two of the deep earthquakes (the Bolivia earthquake and the Russia-China earthquake) have small radiation efficiencies. See color version of this figure at back of this issue.

**Fig. 1.**

---

## Author Comment (AC1) · 18 Jun 2018

**Reply to reviewer 1**

E. Lorenzano and M. Dragoni

Dipartimento di Fisica e Astronomia, Alma Mater Studiorum Università di Bologna, Viale Carlo Berti Pichat 8, 40127 Bologna, Italy

*We answer point-by-point to the reviewer's comments and requests. In the following, figure, page, line and section numbers refer to the Interactive Discussion version of the manuscript.*

General comments:

1) Generally speaking, I find the treatment of the "earthquake" system overly simplified – to the degree that I want to question whether the provided results actually bear any insights into the recurrence of earthquake rupture (including the effect of viscoelastic relaxation). The authors mention that knowledge of the initial state of stress in the

**SED**

system would allow to calculate/predict the following sequence of earthquakes i.e., asperity ruptures (in absence of stress perturbations). While this may be in theory correct, this approach is in my view not appropriate to describe earthquake rupture and recurrence, considering the spatial and temporal variation of physical parameters that in fact control earthquake rupture. In the present work, all that existing and important complexity is removed i.e., not considered.

The present fault model clearly provides a simplified description of real fault dynamics. However, if we aim to a neat understanding of the physics of the seismic source, unnecessary complications must be set apart and different phenomena must be considered separately. As a matter of fact, studying fault dynamics in the framework of a discrete dynamical system represents a tool for enlarging our understanding of the most significant and essential aspects of the seismic activity, such as the stick-slip mechanism governed by the system of forces on the fault. Also, the characterization of the fault as made of a finite number of asperities allows a description by means of a finite number of degrees of freedom, thus making the retrieval of the analytical solutions of the evolution equations possible (section 3). In this analytical framework,

the different phases of the evolution of the system and their distinctive features can be studied by means of a geometrical approach, calculating the orbit of the system in the state-space (section 4 and 6). Of course, taking several physical parameters and their spatial and temporal variation into account is important, but it would require a characterization by means of a model based on continuum mechanics or a numerical approach, which would make it difficult to highlight the basic mechanisms of fault dynamics.

As the reviewer pointed out, several geophysical phenomena are not taken into account in the present model. Some of them (e.g. the interaction between mechanically different regions on a fault, the role of asperity size, the interplay between external stress perturbations and viscoelastic relaxation on a fault) have been object of previous works in the framework of discrete fault models (e.g. Dragoni and Lorenzano, 2017; Lorenzano and Dragoni, 2018a,b) and could be introduced in the present model at the price of increased complication. However, this would go beyond the scope of the present work, which is further explained in the next reply.

2) The authors mention that the aim of the presented study is to expand on previous work (P2L19). But that is not really motivating anything. What are the authors actually trying to constrain/identify? How can the results applied? What insights regarding earthquake rupture does it provide? The study needs an improved motivation/introduction section.

In the present work, we consider a two-asperity fault in the presence of viscoelastic relaxation and provide a more detailed characterization of its dynamics with respect to previous studies (Amendola and Dragoni, 2013; Dragoni and Lorenzano, 2015). First of all, the radiation of elastic waves during seismic events is included, thus presenting a more complete and general solution to the equations of motion (section 3). Afterwards, we show how the particular sequence of slip episodes during a seismic event is controlled by the state of stress on the fault, both at the onset of the event itself (section 4.1) and at the beginning of the interseismic interval preceding the event (section 4.2). In particular, additional constraints with respect to previous works are determined using the condition for the consecutive, but separate, slips of the asperities as a discriminant factor. Then, we focus on these kinds of seismic events

and investigate their dependence on the seismic efficiency of the fault, the intensity of asperity coupling and asperity relative frictional strengths (section 6).

The possible insights on earthquake rupture the model can provide have been discussed by Dragoni and Lorenzano (2015), who also presented an application to the 1964 Alaska earthquake. The authors showed that a major role in this sense is played by the source time function associated with an earthquake. In fact, the number and the amplitudes of humps in a source time function are directly related with the number and sequence of slip episodes during the associated seismic event. An example is shown in Fig. 7 and Fig. 8 for one-mode events 10 and 01, respectively. In turn, the observation of the source function of a seismic event allows to set constraints on the (otherwise unknown) state of stress of the fault that caused it (section 4).

3) The proximity of the two asperities considered here relative to each other should play a role (on the probability of respective rupture modes) – maybe I missed it, but do the authors consider that point?

The proximity of the two asperities is one of the factors determining the value of the parameter $\alpha$, which controls the intensity of coupling and, consequently, the stress transfer between the asperities (Eq. 1). In fact, by comparison with a model based on continuum mechanics, the specific value of $\alpha$ can be estimated as (Lorenzano and Dragoni, 2018)

$$\alpha = \frac{Avs}{2\dot{e}} \tag{1}$$

where $A$ is the area of the asperities, $v$ is the velocity of the tectonic plates, $s$ is the tangential traction (per unit moment) imposed on one asperity by the slip of the other and $\dot{e}$ is the tangential strain rate on the fault due to tectonic loading. For nonoverlapping asperities, the traction produced by point-like dislocations is a good approximation for $s$ (e.g. Dragoni and Lorenzano, 2016). Specifically, we have

$$s = \frac{5}{12\pi} a^{-3} \tag{2}$$

for strike-slip faulting and

$$s = \frac{1}{6\pi} a^{-3} \tag{3}$$

for dip-slip faulting, where $a$ is the distance between the centroids of the asperities. We conclude that the strength of coupling between the two asperities is inversely

proportional to their distance.

The value of $\alpha$ influences several aspects of the dynamics of a seismic event, as predicted by the model. First of all, as shown in Appendix B, it contributes to define the subsets of the state space discussed in section 4, thus controlling the sequence of slip modes in a seismic event. Also, it determines the intensity of static stress drops on the asperities (section 5, Table 1). Finally, it governs the possible sequence of alternate slips of the asperities in a seismic event (section 6.3).

Specific comments:

1) P1L4 – colon after "degrees of freedom" indicates that a list of those is following – but that does not seem to be the case; please rephrase.

In the present model, the state of the fault is characterized by three state variables: the slip deficit of asperity 1, the slip deficit of asperity 2 and the temporal variation of the difference between the slip deficits due to viscoelastic relaxation. Accordingly,

the system has three degrees of freedom, corresponding to the aforementioned state variables. We shall rephrase the Abstract in order to better explain the correspondence between the state variables and the degrees of freedom of the system.

2) P1L5 – the slipping modes should be mentioned here; current formulation too implicit/vague.

We shall rephrase in order to illustrate the difference between the three slipping modes.

3) Abstract – does not stand alone; the reader learns to some extend what the authors wanted to do but not what they learned/have found out; this needs to be included into the abstract.

We shall include the main results of our study in the Abstract.

4) P1L13 – replace "by asperity models": the "by" is wrong, the models don't investigate anything.

The sentence shall be rephrased.

5) P1L22 – how are "non-asperities" defined/characterized? Needs to be mentioned here; they also have a role within the earthquake system and the authors need to state what that role is; include corresponding explanation in the model formulation section.

Asperities on a fault are defined as "unstable" or "strong" regions: they remain locked for most of the time and eventually undergo a sudden failure, catastrophically releasing the deformation energy stored in the medium with the emission of elastic waves. From a frictional point of view, they are characterized as velocity-weakening (VW) regions.

However, faults can accommodate tectonic motion in another way. This second mechanical behaviour is ascribed to "stable" or "weak" fault regions, which exhibit a slow, quasi-static creep during interseismic intervals and afterslip during post-seismic intervals. From a frictional point of view, they are characterized as velocity-strengthening

(VS) regions. In the present work, we neglect the possible presence of such "non-asperities" on the fault. In the framework of a discrete fault model, this problem has been discussed by Dragoni and Lorenzano (2017), who considered a fault containing an asperity and a weak region. The authors suggested a value of 0.1 for the ratio between the steady-state frictional stress of the weak region and the static frictional stress of the asperity: in fact, asperity models assume that weak regions may slip at a much lower stress level than asperities. By combining elements of the present model with the model of Dragoni and Lorenzano (2017), it would be possible to study the interaction between seismic slip, afterslip and viscoelastic relaxation; however, this kind of analysis is beyond the scope of the present work (see reply to general comment #2).

6) P2L17 – what the authors mean with "source functions"? Is that source time function? Please clarify.

We call "source function" the rate of release of seismic moment as a function of time, that is, the moment rate function. For the sake of clarity, the expression "source

function" shall be replaced with "moment rate function" throughout the manuscript.

7) P2L24 – the term "seismic efficiency" should be defined properly.

The seismic efficiency of the fault is defined as the ratio between the energy radiated as seismic waves and the total elastic energy released by a dislocation on the fault. We shall add this definition to the revised version of the manuscript.

8) P3L3 – language is vague "by a much higher friction than the surrounding region of the fault": be specific/quantitative please.

See reply to specific comment #5.

9) P3L4 – I cannot follow that logic: the authors argue that they can neglect the seismic moment contributed from the "weaker regions" of the fault that surround the asperities, but why? Regardless of strength, if the fault slips (coseismically) then it will contribute

to seismic moment, so I want to question the author's approach here; they need to better explain justify this simplification.

As a matter of fact, the fault region surrounding the asperities does give a contribution to the coseismic seismic moment release. However, one of the basic assumptions of asperity models is that the bulk of seismic moment release in a seismic event is ascribed to asperity slip, corresponding to the largest humps in the source time function of the event itself. A possible way to account for the contribution of the weaker fault region has been presented in the framework of a two-asperity discrete fault model by Dragoni and Santini (2015); the authors applied their model to the 1964 Alaska earthquake, showing that the slip of the weaker fault region contributed only to about 20% of the overall moment release associated with that event. Although taking this aspect into account would result in a better fit between the observed and modelled source time functions of an event, the conclusions of the theoretical study presented here would not be affected.

10) P3L20 – why using a rate-dependent law? Did the authors experiment with other

laws as well? Please better motivate the use of this friction law.

The purpose of the present work is to provide a macroscopic characterization of the mechanics of the seismic source, neglecting a detailed description of stress, slip and friction distribution on the fault. Accordingly, it is sufficient to replicate the typical stick-slip behaviour of the fault, a result that can be properly achieved by adopting the simplest formulation of a rate-dependent friction law, corresponding to a constant static friction threshold and a constant dynamic friction. The use of more accurate descriptions of frictional resistance such as the rate- and state-dependent friction laws (Ruina, 1983; Dieterich, 1994) would ony result in a more complex modelling and provide negligible improvements to our conclusions.

11) P12L8 – the authors will need to explain how their analytical toy model is able to inform our understanding of earthquake rupture and rupture sequences; after all, we don't know the "initial state of stress" and real faults do exhibit stress perturbations, along with a range of other processes and parameters that affect earthquake rupture and that are not considered here. So, how does the presented study help to learn

about earthquakes?

See reply to general comment #1.

**References**

Amendola, A. and Dragoni, M. (2013). Dynamics of a two-fault system with viscoelastic coupling, Nonlin. Processes Geophys., 20, 1 – 10, doi:10.5194/npg-20-1-2013.

Dieterich, J. (1994). A constitutive law for rate of earthquake production and its application to earthquake clustering. J. Geophys. Res., 99, 2601 – 2618, doi:10.1029/93JB02581.

Dragoni, M. and Lorenzano, E. (2015). Stress states and moment rates of a two-asperity fault in the presence of viscoelastic relaxation. Nonlinear Process. Geophys., 22(3), 349 – 359, doi:10.5194/npg-22-349-2015.

**SED**

Dragoni, M. and Lorenzano, E. (2016). Conditions for the occurrence of seismic sequences in a fault system. Nonlin. Process. Geophys., 23(6), 419 − 433, doi:10.5194/npg-23-419-2016.

Dragoni, M. and Lorenzano, E. (2017). Dynamics of a fault model with two mechanically different regions. Earth Planets Space, 69(145), doi:10.1186/s40623-017-0731-2.

Dragoni, M. and Santini, S. (2015). A two-asperity fault model with wave radiation. Phys. Earth. Planet. In., 248, 83  93, doi:10.1016/j.pepi.2015.08.001.

Lorenzano, E. and Dragoni, M. (2018): A fault model with two asperities of different areas and strengths, Math.Geosci., https://doi.org/10.1007/s11004-018-9738-x.

Lorenzano, E. and Dragoni, M. (2018): Complex interplay between stress perturbations and viscoelastic relaxation in a two-asperity fault model, Nonlin. Process.

Geophys., 25, 251-265, https://doi.org/10.5194/npg-25-251-2018.

Ruina, A. (1983). Slip instability and state variable friction laws. J. Geophys. Res., 88, 10359 – 10370, doi:10.1029/JB088iB12p10359.

---

## Editor Comment (EC1) · F. Rossetti (Editor) · 24 Jun 2018

Dear Dr. Lorenzano,

based on the reviewers' reports and my own assessment of the review process, my evaluation is your manuscript needs re-thinking and re-organisation.

The main issue regards the relevance of the study to the "real earthquake" system. Based on the audience of the Journal, it is necessary to better introduce and discuss limitation of the scientific rationale adopted in the study and try to finger out possible insights and implications on the understanding of the earthquake source region. Furthermore, I noted a series of recent papers with your authorship with similar and somehow overlapping topics with those presented in your submitted manuscript. It is

therefore compulsory that all background information (including other relevant literature in the field) is clearly referenced and the scope and motivation of the study better detailed in the revised manuscript: which the expected advancement of knowledge with respect to the state-of-the art?

To conclude, broader impact of the proposed research is thus largely (and critically) depending on how the obtained results can be extrapolated to improve our knowledge on the main physical properties that control the fault mechanics and the earthquake cycle (rupture and recurrence) in general.

Sincerely, Federico Rossetti

---

## Author Comment (AC2) · 24 Jun 2018

**Reply to reviewer 2**

E. Lorenzano and M. Dragoni

Dipartimento di Fisica e Astronomia, Alma Mater Studiorum Università di Bologna, Viale Carlo Berti Pichat 8, 40127 Bologna, Italy

*We answer point-by-point to the reviewer's comments and requests. In the following, equation, figure, page, line and section numbers refer to the Interactive Discussion version of the manuscript.*

1) Previous models have studied fully dynamic earthquake cycles on a fault with asperities [e.g., Lui and Lapusta, "Repeating microearthquake sequences interact predominantly through postseismic slip", Nature Communications, 2016]. A review of these previous studies is lacking in the introduction section. In particular, how can the modeling approach in this paper contribute to our understanding of earthquake cycles?
Following the referee's recommendation, we shall add in the Introduction a short review of previous papers studying fault slip in the presence of friction heterogeneities, in particular Mikumo and Miyatake (1995), Beroza and Mikumo (1996), Somerville et al. (1999), Pisarenko (2002), Mai and Beroza (2002), Johnson (2010), Lui and Lapusta (2016) and Zielke et al. (2017).

As to the contribution of the present paper to our understanding of seismic cycles, one of the merits of discrete fault models is to allow a systematic study of the evolution of the system as a function of initial conditions. This is possible because the evolution can be represented as an orbit in a low-dimensional state space, which can be continued for an arbitrary number of cycles. Examples of such studies have been given by Dragoni and Santini (2010, 2012) and Dragoni and Piombo (2015). However, in the present paper we focus on single events, showing how the initial conditions control the sequence of slipping modes during the event.

2) On Page 3, the asperity is characterized by a much higher friction than the surrounding region, which I don't think is necessarily true for a real fault. Could the authors provide some observations that support this view?

Friction on faults is of course a continuous function of position. However, it has been recognized that, when an earthquake occurs, most of the seismic moment is released by a small number of regions where friction is much higher than in the rest of the fault: this observation is at the basis of asperity models (e.g. Scholz, 1990). In these models, the fault surface is separated into two kinds of regions, characterized by high and low friction, respectively. It is assumed that a higher friction corresponds to a higher accumulated stress and to a larger slip (Somerville et al., 1999).

3) The model assumes a rate-dependent friction law instead of a rate and state dependent friction law that is observed in laboratory experiments and used in fully dynamic earthquake cycle models. The authors replied to the other reviewer that using rate and state dependent friction laws would "provide negligible improvements to our conclusions". However, if the friction changes over time as defined by the state variable, it will significantly affect the recurrence intervals of seismic events.

As noticed by the referee, our model does not include a dependence of friction on the

state of the fault. This might imply a slow change of static friction with time, when the fault is at rest, and a change in the duration of interseismic intervals. This effect could be easily incorporated in the model by introducing a dependence of the parameters $\beta$ and $\epsilon$ on time. However, we neglect this further complication at the present stage. This assumption of the model will be acknowledged in the revised version of the paper.

4) On Page 5, the authors mentioned that they consider the case of underdamping because seismic efficiency of faults is small. I don't think this is true. Radiation efficiency depends on the earthquake type and is not always small. I've attached Figure 8 from Venkataramen and Kanamori [2004]. For example, the radiation efficiency of tsunami earthquakes is usually lower than other types of earthquakes.

The radiation efficiency used by Venkataraman and Kanamori (2004) is not the seismic efficiency mentioned in our paper: the two quantities have different definitions. As shown by Kanamori (2001), the radiation efficiency is always greater than the seismic efficiency. Assuming the overdamped solution allows seismic efficiencies as large as 0.33 (Dragoni and Santini, 2017), corresponding to greater values of radiation

efficiency, in agreement with observations.

5) It's hard to relate the proposed models to realistic cases. In section 4.1, the authors discussed the different earthquake models when Pk belongs to different segments on the face AECD. If we picked a region, e.g., Parkfield, how could we determine which segment it belongs to?

In section 4.1, the relationship between the stress state of the fault at the onset of a seismic event (corresponding to a point $P_k$ in the state space) and the particular sequence of slipping modes in the event itself is discussed. For instance, if the state $P_k$ belongs to the trapezoid $\mathbf{Q_1}$ on the face $AECD$ of the sticking region, the seismic event will be due to the sole slip of asperity 1 (i.e., one-mode event 10).

Different faults do not correspond to different subsets of the sticking region. If we focused on a given fault, the proper set of parameters for that specific fault (that is, the values of $\alpha, \beta, \gamma, \epsilon, \Theta$ and $V$) could be retrieved by exploiting inter-, co- and post-seismic field observations. In turn, the sticking region $\mathbf{H}$ corresponding to that

particular fault could be determined and its geometric features could be studied as in section 4 and Appendix B. Once completed this preliminary analysis, one could associate an observed seismic event on the fault with a specific sequence of slipping modes (using the moment rate function of that event as a constraint, as explained in the reply to reviewer 1) and thus gain information on the stress state that originated the event. An exemplification of such procedure can be found in the work of Lorenzano and Dragoni (2018a), who modelled the Landers fault as a two-asperity fault and showed that the 1992 earthquake that took place on that fault could have been originated by any of the states belonging to the segment $s_2$ of the face $BCDF$ of $\mathbf{H}$.

6) Fig.7 and Fig. 8 are not cited in the manuscript. Though the peak moment rate amplitudes are slightly different in the figures, the moment rate functions have very similar shapes for events 10 and 01. Why is that?

Figures 7 and 8 are cited at page 15, line 7. They show both the slip amplitude and the moment rate function of one-mode events 10 and 01, respectively. The features of the moment rate functions are a direct consequence of the solutions to the equations

of motion from which they are calculated, according to Eq. (78). Specifically, the model predicts that events starting from a global stick phase and associated with the slip of a single asperity have the same duration and the same shape for the associated slip amplitude, as shown by Eqs. (42), (45), (61) and (63). As the reviewer pointed out, the only difference between the moment rate functions of the two events is their peak value: in fact, the final slip amplitude (63) of asperity 2 differ by a factor $\beta$ from the final slip amplitude (45) of asperity 1, with $0 < \beta < 1$.

The aforementioned characteristics depend on the assumptions of asperities with equal areas and same radiation damping (section 2). These hypotheses have been relaxed by Lorenzano and Dragoni (2018b), who investigated how the difference between the asperity areas affects several aspects of the seismic events generated by a two-asperity fault.

**References**

Beroza, G. C. and Mikumo, T., 1996. Short slip duration in dynamic rupture in the

presence of heterogeneous fault properties. J. Geophys. Res. 101, 22449 – 22460.

Dragoni, M. and Piombo, A. (2015). Effect of stress perturbations on the dynamics of a complex fault, Pure Appl. Geophys., 172 (10), 2571 – 2583.

Dragoni, M. and Santini, S. (2010). Simulation of the long-term behaviour of a fault with two asperities, Nonlin. Processes Geophys., 17, 777–784, doi:10.5194/npg-17-777-2010.

Dragoni, M. and Santini, S. (2012). Long-term dynamics of a fault with two asperities of different strengths, Geophys. J. Int., 191, 1457–1467.

Dragoni, M. and Santini, S. (2017). Effects of fault heterogeneity on seismic energy and spectrum, Phys. Earth Planet. Inter., 273, 11ÂŰ-22.

Johnson, L. R. (2010). An earthquake model with interacting asperities. Geophys. J.

[Figure]

Int. 182: 1339–1373. doi:10.1111/j.1365-246X.2010.04680.x.

Kanamori, H. (2001). Energy budget of earthquakes and seismic efficiency. In: Earthquake Thermodynamics and Phase Transformations in the Earth's Interior, Academic Press, pp. 293–305.

Lorenzano, E. and Dragoni, M. (2018a): Complex interplay between stress perturbations and viscoelastic relaxation in a two-asperity fault model, Nonlin. Process. Geophys., 25, 251-265, https://doi.org/10.5194/npg-25-251-2018.

Lorenzano, E. and Dragoni, M. (2018b): A fault model with two asperities of different areas and strengths, Math.Geosci., https://doi.org/10.1007/s11004-018-9738-x.

Lui, S. K. Y. and Lapusta, N. (2016). Repeating microearthquake sequences interact predominantly through postseismic slip, Nature Communications 7, https://doi.org/10.1038/ncomms13020.

Mai, P. M. and Beroza, G.C. (2002). A spatial random field model to characterize complexity in earthquake slip. J. Geophys. Res. 107, 2308, doi:10.1029/2001JB000588.

Mikumo, T. and Miyatake, T. (1995). Heterogeneous distribution of dynamic stress drop and relative fault strength recovered from the results of waveform inversion: the 1984 Morgan Hill, California, earthquake. Bull. Seismol. Soc. Am. 85, 178–193.

Pisarenko, D. (2002). Elastodynamical mechanism of rate-dependent friction. Geophys. J. Int. 148, 499–505.

Scholz, C. H. (1990). The Mechanics of Earthquakes and Faulting, Cambridge Univ. Press, pp. 439.

Somerville, P., Irikura, K., Graves, R., Sawada, S., Wald, D., Abrahamson, N., Kagawa, T., Iwasaki, Y., Smith, N. and Kowada, A. (1999). Characterizing crustal earthquake slip models for the prediction of strong ground motion, Seismol. Res. Lett., 70, 59–80.

Venkataraman, A. and Kanamori, H. (2004). Observational constraints on the fracture energy of subduction zone earthquakes, J. Geophys. Res., 109, doi: 10.1029/2003JB002549.

Zielke, O., Galis, M. and Mai, P.M. (2017). Fault roughness and strength heterogeneity control earthquake size and stress drop. Geophys. Res. Lett. 44, 777–783, http://dx.doi.org/10.1002/2016GL071700.